# Time-Course Changes in Oxidative Stress and Inflammation in the Retinas of rds Mice: A Retinitis Pigmentosa Model

**DOI:** 10.3390/antiox11101950

**Published:** 2022-09-29

**Authors:** Antolín Cantó, Javier Martínez-González, Inmaculada Almansa, Rosa López-Pedrajas, Vicente Hernández-Rabaza, Teresa Olivar, María Miranda

**Affiliations:** Department of Biomedical Sciences, Faculty of Health Sciences, Institute of Biomedical Sciences, Cardenal Herrera-CEU University, CEU Universities, 46115 Valencia, Spain

**Keywords:** oxidative stress, inflammation, hydroxynonenal, 8-oxoguanine, glutathione, macroglia, microglia

## Abstract

(1) Background: Retinitis pigmentosa (RP) is characterized by progressive photoreceptor death. A Prph2Rd2 or an rds mouse is an RP model that closely reflects human RP. The objective of this study was to investigate the relationship of rod and cone death with oxidative stress and inflammation in rds mice. (2) Methods: The retinas of control and rds mice on postnatal days (PN) 11, 17, 21, 28, 35, and 42 were used. Oxidative damage to macromolecules, glutathione (GSH and GSSG), GSH synthesis enzymes, glial fibrillar acidic protein (GFAP), ionized calcium-binding adapter molecule 1 (Iba1), and cluster of differentiation 68 (CD68) was studied. (3) Results: The time sequence of oxidative stress and inflammation changes in rds mice occurs as follows: (i) At PN11, there is a small increase in photoreceptor death and in the microglial cells; (ii) at PN17, damage to the macromolecules is observed; (iii) at PN21, the maximum photoreceptor death rate is detected and there is an increase in GSH-GSSG and GFAP; (iv) at PN21, the microglial cells are activated; and(v) at PN28, there is a decrease in GSH synthesis enzymes. (4) Conclusions: These findings contribute to the understanding of RP physiopathology and help us to understand whether oxidative stress and inflammation are therapeutic targets. These findings contribute to our understanding that, in RP, oxidative stress and inflammation evolution and their relationship are time-dependent. In this sense, it is important to highlight that both processes are potential therapeutic targets in this disease.

## 1. Introduction

Retinitis pigmentosa (RP) is an inherited retinal degenerative disorder with a global prevalence of 1 in 4000 people. The main features of RP are a loss of visual field, a loss of scotopic vision, intraretinal pigmentation in a specific form called bone spicules, and, finally, complete blindness [1,2]. There are 50 genes and over 3100 different mutations related to non-syndromic RP. Syndromic RP, including Usher syndrome and Bardet-Biedl syndrome, are also highly heterogeneous, and 1200 pathologic mutations have been implicated in these two diseases [3].

RP is characterized by progressive cell death in the retinal outer nuclear layer (ONL), where photoreceptor nuclei are found. First, the rods die because of the mutation. As a result, the cones suffer a progressive size and number reduction [4]. Morphological changes take place even before photoreceptor cell death. This can be observed because there is a reduction in the outer segment layer thickness, which is produced due to the shortening of the outer segments of rods and cones [5,6]. RP progression can be divided into stages. First, the photoreceptors suffer a period of stress. Second, the photoreceptors start dying. Finally, nervous, glial, and vascular remodeling begins [7,8]. Two processes are related to all three phases: oxidative stress and inflammation [9].

Oxidative stress has been described as a “change in the pro-oxidant/antioxidant equilibrium in favor of the former, potentially leading to cell damage” [10]. Under normal physiological circumstances, ocular tissue possesses various intrinsic antioxidants to overwhelm the oxidative stress that occurs due to the normal metabolism of the cells. However, during ocular damage or disease, the overproduction of reactive oxygen species (ROS) and free radicals bypasses normal intrinsic antioxidant mechanisms, resulting in oxidative stress [11]. This stress modifies biological molecules such as proteins, nucleic acids, and lipids, which can contribute to disease development.

Several antioxidant drugs, such as N-acetylcysteine, lipoic acid, vitamin E, vitamin C, and superoxide dismutase mimetics, have been tested and show an important beneficial effect on photoreceptor survival in mice models [9]. In addition, genetically modified mice with an overexpression of superoxide dismutase 1 (SOD1) and glutathione peroxidase 4 (GPx4), which are enzymes involved in oxidative stress metabolism, demonstrate a neuroprotective effect in photoreceptor cells [9]. However, contradictory results have been obtained regarding different antioxidant interventions, especially in human trials. Therefore, it is crucial to precisely quantify oxidative stress to establish its exact function in RP.

The direct measurement of reactive species (RS) in animal or cellular models of RP is one method of determining oxidative stress. The direct determination of RS is difficult to achieve because these molecules have a short lifespan and react quickly with other cellular components [12]. Another possibility for determining oxidative stress is to estimate the damage that this RS causes; the result of the reaction of RS with a biological molecule may then be used as a biomarker of oxidative stress [13]. In this sense, lipid peroxidation is a complex process [12] that has been used as an index of oxidative stress in cell membranes [13]. Aldehydes, including malondialdehyde (MDA) and 4-hydroxynonenal (HNE), are examples of end products of fatty acid peroxidation that are often used to evaluate oxidative stress [13]. 8-Hydroxy-2′-deoxyguanosine (8-OHdG) is one of the main oxidative modifications in DNA subjected to attack by hydroxyl radicals and has been studied extensively to assess oxidative stress in several sample types [12]. Finally, the determination of the total antioxidant status or of different natural antioxidants can also be correlated with the extent of oxidative stress [12]. One example of a natural antioxidant that can be easily examined is glutathione (GSH), the major intracellular antioxidant.

There is no single method recommended as the gold standard for evaluating oxidative stress situations. One individual marker that only partly reflects the oxidative grade would be insufficient. Instead, it is preferable to use integrative approaches and select different methods that may give us a more accurate idea of the exact oxidant/antioxidant balance in cases of retinal degeneration and other disease types.

Inflammation is considered a hallmark of numerous chronic degenerative pathologies [14]. Similar to other retinal pathologies, macro- and microglial reactivity induced by inflammation plays an essential role in RP [15,16].

Müller cells and astrocytes are the two retinal macroglia cells. The main role of Müller cells is to provide structural and trophic support to all other retinal cells. Gliosis is the response of macroglia to pathogenic stimuli to restore the normal state of the tissue [17]. Initially, gliosis shows a protective effect, but, if gliosis becomes chronic, it can also contribute to retinal impairment by releasing cytotoxic factors, via remodeling processes, and because it participates in glial scar formation [18]. Most gliosis is related to Müller cells and can be characterized by glial fibrillar acidic protein (GFAP) upregulation [19,20].

Retinal microglia, which are the resident macrophages of this tissue, show a detrimental role in cell death [19]. Microglia activation increases the release of pro-inflammatory and pro-phagocytosis factors, such as TNF-α and IL-1β, which lead to cellular death and more microglia activation, setting up a positive feedback effect [21]. It has been shown in pharmacological experiments in RP murine models that the inhibition of microglia activity leads to a neuroprotective effect in the retina [21]. Furthermore, genetically modified mice without microglia expression showed an amelioration of photoreceptor degeneration [21]. Increases in microglia activity can be observed and quantified by two means: the migration of microglia cells through the different retinal layers, and morphological changes of microglial cells [18]. A better understanding of the dual, neuroprotective, and cytotoxic effects of macro- and microglial involvement in retinal pathologies would assist in finding new target therapies for these diseases.

Oxidative stress and inflammation have been widely studied in two classical RP mouse models: rd1 and rd10 mice [22,23,24,25,26,27]. Prph2Rd2, also called the rds mouse model, is the model used in the present study. Peripherin 2 (Prph2) is a photoreceptor-specific glycoprotein that is needed for the correct production of rod and cone photoreceptor outer segments [28]. The human Prph2 gene has been correlated with 151 individual disease-causing mutations with extremely unpredictable patient diagnoses, extending from RP to macular degeneration [29]. Rds mice show a slow progression of RP disease, which starts on postnatal (PN) day 14 and continues until month 9, when the ONL has disappeared [30]. The rate of cell death is fast until month 2 or 3, when the ONL is reduced to half of its original thickness. Afterward, cell death diminishes, and retinal degeneration progresses slowly [30]. At month 9, the peripheral parts of the retina are deficient in visual cells, and at 12 months, the entire retina is affected [30]. In the central retina, rod and cone cells are similarly affected up to the 6-month stage. In the peripheral and central retina at 9 months, an increase in the occurrence of cones is recorded, indicating augmented vulnerability of the rods. In the rds retina, increased incidence and stainability of macrophages are visible in the inner retina at 11 days. Macrophages migrate and are observed in the ONL during degeneration [30]. The degree of cell death in rds mice closely reflects the rate observed in human RP, more so than in other types of retinal degeneration in other animal models [31].

The sequential relationship of rod and cone death with oxidative stress and inflammation biomarkers has not been fully elucidated in the retinas of rds mice. Therefore, the main objective of this study was to investigate the expression status of a wide range of oxidative stress and inflammation biomarkers (i.e., HNE, nuclear oxidative damage, GSH metabolism, GFAP, ionized calcium-binding adapter molecule 1 (Iba1), and cluster of differentiation 68 (CD68)) during the development of retinal degeneration in rds mice.

## 2. Materials and Methods

Control (C3Sn.BLiA-Pde6b^+^/DnJ, homozygous for Pde6b^+^) and rds mice (C3A.CG-Pde6b^+^ Prph2^Rd2^/J, homozygous for Prph2^Rd2^) were used in this work. The mice were derived from the Jackson Laboratory colony (The Jackson Labs, Bar Harbor, ME, USA). The animals were harbored in the facilities of the Research Unit of CEU Cardenal Herrera University (Valencia, Spain). They were maintained in cages under regulated conditions of humidity (60%) and temperature (20 °C). The animals were housed under regular cyclic lighting (12 h) and had free access to water and a normal diet (Harlan Ibérica S.L. (Barcelona, Spain)). Light illuminance was provided by cool-light LED bulbs, and the irradiance at the cage level was found to be between 6.22 and 9.32 µW/cm^2^.

The control and care of the mice were authorized by the CEU Cardenal Herrera Universities Committee for Animal Experiments (approval code 2020/VSC/PEA/0094) and we completed the Association for Research in Vision and Ophthalmology (ARVO) Statement for the Use of Animals in Ophthalmic and Vision Research.

The day of birth was considered to be postnatal day 0 (PN0). Twelve experimental groups were used: control and rds mice at PN11, PN17, PN21, PN28, PN35, and PN42 (four mice in each group were used for the histological and immunohistochemistry studies and four mice per group were used for Western blot analysis). These postnatal days were selected because: (1) it has been reported that initial changes in macrophages are visible in the inner retina at PN11 in rds retinas; (2) the photoreceptor peak of cell death in rds retinas occurs around PN21; (3) PN17 was considered interesting because it is an intermediate state between the first changes and the peak of cell death; (4) in rds retinas, there is an accelerated cell death period that lasts until PN28: (5) a slower cell death rate period in rds photoreceptors starts around PN35; and (6) PN42 was selected to monitor what happens during the slow photoreceptor death period [30,31].

### 2.1. Histological and Immunofluorescence Studies

The mice were euthanized via neck dislocation. The eyes were enucleated, the cornea of each eye was pierced with a needle, and then, the eye was immediately embedded in paraformaldehyde (4% PFA) for 2 h. They were then washed three times for 10 min with 0.1 M phosphate buffer saline (PBS). Ultimately, the eyes were cryoprotected using increased sucrose–PBS solutions (10–30%). They were placed in Tissue Tek (Sakura Europe, Barcelona, Spain), and retinal portions of 8 µm were taken with the help of a cryostat (Leica CM 1850 UV Ag protect, Leica Microsistemas S.L.U., Barcelona, Spain). The sections were placed on superfrost slides (Thermo Fisher Scientific, Braunschweig, Germany) and kept at −20 °C.

#### 2.1.1. Hematoxylin–Eosin Assay

The eye sections were covered with a drop of Harris hematoxylin (Química Clínica Aplicada, Tarragona, Spain) for 3 min. They were then washed with tap water for 5 min, covered with eosin (Química Clínica Aplicada, Tarragona, Spain) for 1 min, and rinsed with tap water for 5 min. Next, the eye slices were dehydrated with increasing alcohol solutions: 70% (1 min), 96% (2 min), and 100% (2 baths of 1 min). The tissue sections were washed twice (for 5 and 2 min) in xylene (WWR International, Fontenay Sous Bois, France). Finally, the sections were mounted on a coverslip on Diamount medium (Diapath, Martinengo, Italy).

The sections were then used for morphological analysis, including determinations of the number of rows on the ONL. The number of rows in the ONL was measured manually in three different sections of the retina: the far periphery (near the ora serrata), midperiphery (between the ora serrata and the optic nerve), and nerve (just beside the optic nerve) (the exact localization and distance of the different areas can be observed in Appendix A).

#### 2.1.2. Terminal Deoxynucleotidyl Transferase (TUNEL) Assay

To detect dying cells, a terminal deoxynucleotidyl transferase (TUNEL) assay was carried out using an in situ detection kit, as stated in previous research (Roche Diagnostics, Mannheim, Germany) [31]. Retinal images were taken using a Leica DM 2000 microscope attached to a Nikon DS-Fi1 camera. We used the Leica application Suite version 2.7.0 R1 (Leica Microsystems SLU, Barcelona, Spain) program. The computer software ImageJ Fiji version 1.52p (Bethesda MD, USA) was used to quantify the number of TUNEL-positive cells in the three areas described above.

#### 2.1.3. Retinal Immunohistochemistry

The tissue sections were rehydrated for 15 min with PBS. Amounts of 5% normal goat serum, PBS-BSA 1%, and Triton 0.3% for 1 h at 4 °C, were used to block nonspecific binding sites. Subsequently, the sections were rinsed three times for 10 min with PBS-Triton 0.3%, and the primary antibodies (listed in Table 1) were incubated with PBS-Triton 0.3% during the night at 4 °C. The sections were rinsed three times for 10 min with PBS-Triton 0.3%, and the secondary antibody, Alexa 488 (Invitrogen, Life Technologies, Madrid, Spain), was incubated for 1 h at 4 °C (except for avidin staining). Lastly, the slices were mounted using Vectashield mounting medium with DAPI (Vector, Burlingame, CA, USA). Fluorescence microscopy was performed using a Nikon DS-Fi1 camera joined to a Leica DM 2000 microscope. The Leica Application Suite version 2.7.0 R1 program (Leica Microsystems SLU, Barcelona, Spain) was used.

Changes in the rod cell numbers, reduced and oxidized glutathione (GSH-GSSG) expression, and macroglial reaction (which is labeled by the marker GFAP) where evaluated using the percentage of area occupied by these proteins. The cone cell numbers, avidin-positive cells, microglial changes (Iba1), and macrophage presence (CD68), were studied by counting the number of positive cells per unit area. To determine hydroxynonenal expression (HNE), we quantified the green pixel intensity.

Cone photoreceptor numbers were counted according to the pedicles at the base of the cone and to the nucleus, which was stained with a lighter green compared to the cell body.

Rod cells, cone cells, HNE, and GSH-GSSG were quantified only in the section close to the optic nerve because its expression is similar in all the retinal sections. Avidin, GFAP, Iba1, and CD68 were quantified in the far periphery, midperiphery, and nerve sections. HNE was quantified in the in six different layers of the retina: the ganglion cell layer (GCL), inner plexiform layer (IPL), inner nuclear layer (INL), outer plexiform layer (OPL), ONL, and segment layer. GSH was quantified in the GCL, IPL, INL, and OPL (no or very low expression was found in the ONL and SL), and finally, Iba1 was quantified in the GCL, IPL, INL, OPL, and ONL.

The photos used for quantifications were taken at 20X magnification and were analyzed using the software ImageJ Fiji version 1.52p (the photos used in the figure of this manuscript were taken at 40X magnification).

Number of TUNEL, avidin, and CD68-positive cells were counted per unit area, and the entire retina was used for these quantifications. Iba1-positive cells were counted per unit of area in five different retinal layers (GCL, IPL, INL, OPL, and ONL). The Iba1 branch number and length were analyzed using the above-mentioned software, ImageJ, which is able to take an image and draw a line skeleton inside the image, allowing us to measure the number of branches per cell body and the length of these branches.

GSH-GSSG was estimated by determining the occupied percentage area of the staining in four retinal layers (GCL, IPL, INL, OPL, and ONL), but previously, a threshold was established. GFAP was measured similarly to GSH-GSSG, and the occupied percentage area of the staining was estimated in the entire retina. The GSH-GSSG and GFAP photographs were always performed in the same conditions of illumination and intensity.

Finally, the quantification of HNE was performed by studying the intensity of the staining in the following retinal layers: GCL, IPL, INL, OPL, ONL, and SL.

### 2.2. Western Blot Analysis

A total of 50 μL of radioimmunoprecipitation (RIPA) buffer was used to homogenize both retinas of each mouse (four mice per group were used). The samples were then centrifuged at 13,000 rpm for 10 min at 4 °C. The supernatant was compiled, and protein concentration was obtained using the Bradford technique [32]. A total of 50 μg of protein was run for 2 h on 4–20% acrylamide:bisacrylamide gels at 100 V. The proteins were shifted to membranes of nitrocellulose (Amersham^TM^ Hybond ECL; GE Healthcare Life Sciences, Barcelona, Spain) and blocked with 0.01 M PBS-Tween 20 0.1%, with 5% *w*/*v* non-fat milk, for 1 h.

The membranes were used with the antibodies listed in Table 2 (glutamate–cysteine ligase catalytic subunit (GCLC), glutamate–cysteine ligase modifier subunit (GCLM), and oxoglutarate carrier (OGC)). The membranes were incubated at 4 °C overnight, and the primary antibodies was distinguished using a horseradish peroxidase-coupled secondary anti-rabbit antibody (F (ab′) 2–HRP, goat anti-rabbit). The signal was detected using an enhanced chemiluminescence developing (ECL) kit (Amersham Biosciences, Buckinghamshire, UK) and measured using densitometry (Image Quant™TL, GE Healthcare Life Sciences, Barcelona, Spain).

### 2.3. Statistical Analysis

Statistical analysis was carried out using IBM SPSS version 25. The normality of the population was ensured using a Shapiro-Wilk test, and variance homogeneity using Levene’s test. Finally, a two-way ANOVA was performed. To understand which groups were different, the Bonferroni test was used. Significance was set at *p* < 0.05.

In the main figures of the manuscript, you can find the difference between the control and the rds group at each age and the difference between the different ages of the rds group. Due to the enormous amount of data obtained when performing the two-way ANOVA, it was decided not to add symbols that show the differences between the different ages of the control group to the figures that are shown in Appendix A.

## 3. Results

### 3.1. Morphological Characterization of Retinal rds Degeneration

Although photoreceptor degeneration in the retinas of rds mice with histological analysis of the ONL has been widely studied [30,33], it is also well known that light intensity is an essential element in the retinal degeneration development of different RP animal models [34]. Therefore, we performed a detailed analysis of the morphometric changes and photoreceptor death in our housing and lighting conditions to ensure an accurate interpretation of all our experimental results.

Figure 1A,B show hematoxylin–eosin sections of the retinas from the two mouse groups used (control and rds mice) at PN11, PN17, PN21, PN28, PN35, and PN42. Ocular sections through the far periphery (near the ora serrata), midperiphery (between the ora serrata and the optic nerve), and nerve (just beside the optic nerve) were used. At PN11 and PN17, no differences between the number of rows in the ONL in the retinas of the control and rds mice were observed. At PN21, PN28, PN35, and PN42, the number of photoreceptor rows in the ONL in all the retinal areas studied decreased progressively in the rds mice compared to the control mice. This progressive decrease in the number of photoreceptors in the rds mice was recognized because the number of row cells in the ONL at PN11 and PN17 was greater than at PN21, PN27, PN35, and PN42 in all the retinal areas studied. Similarly, the number of rows in the ONL was significantly higher at PN21 than at PN28, PN35, and PN42. Finally, the number of rows of photoreceptors was higher at PN28 than at PN35 and PN42. In line with other studies, these results confirm that the photoreceptor peak of cell death occurs around the third postnatal week (PN21) and that there is an accelerated cell death period that lasts until PN28, followed by a slower cell death rate period (that starts around PN35) [30,34].

A TUNEL assay was conducted to further characterize the period of photoreceptor cell death in the rds mice. Our results (Figure 2A,B) show that cellular death was significantly higher in the rds retinas than in the control retinas at every studied age and area, apart from PN42 (at PN42, differences were found only in the total retina). Although the number of TUNEL-positive cells was significantly greater in the rds retinas at PN11 and PN17 than in the control retinas (meaning that the death process starts very early in rds retinas), we observed that the number of dying cells in the ONL increased markedly at PN21 in the rds retinas compared to the control retinas. At PN28 and PN35, retinal cell death declined minimally in the rds mice, and no significant difference was found between the number of TUNEL-positive cells in the rds and control retinas at PN42. The previously mentioned findings are in line with prior research on the existence of an accelerated cell death period followed by a slower cell death period.

When studying the area stained with rhodopsin, we observed an increase in this rod marker in the control mice from PN11 to PN21, probably because the retina was not completely developed until PN21. When observing rod staining in the rds retinas, a similar result was observed, and the rods increased from PN11 to PN21 and PN28 (Figure 3A). After PN28, the rods began to decrease again until PN42. Moreover, the rds mice had significantly fewer rods than the control mice at all the postnatal ages studied.

The number of positive cells for cone staining remained constant from PN11 to PN42 in the control retinas but decreased in the rds mice during retinal development. The number of cones was significantly greater at PN11 than at PN28, PN35, and PN42 (Figure 3A). In addition, a significantly lower number of cones was observed in the rds retinas than in the control retinas on all the postnatal days studied, and by more than 50% at PN42 (Figure 3A).

### 3.2. Oxidative Stress Characterization in rds Mice

#### 3.2.1. Higher Lipid Peroxidation in rds Retinas

HNE staining was employed as a marker for oxidative injury, particularly lipid peroxidation. Immunostaining for HNE was performed on the retinal sections of the rds and control mice. Our results showed HNE staining in all the retinal layers in both the rds and control mice (Figure 4A,B). We determined the intensity of this mark in six different layers of the retina: GCL, IPL, INL, OPL, ONL, and SL. The lowest staining intensity was observed in the ONL (Figure 4A).

No differences were monitored in HNE staining between the rds and control mice at PN11 in the GCL, IPL, INL, and OPL. However, HNE staining was more than 1.5 times higher in the rds retinas than in the control retinas in the GCL, IPL, INL, and OPL at PN17, PN21, PN28, PN35, and PN42. Interestingly, this oxidative stress marker began to increase at PN17, before the peak of cell death, and remained elevated until PN42.

HNE staining was slightly different in the ONL of the rds mice. The rds retinas showed significantly lower HNE expression than the control mice at PN11 in the ONL, but no differences between groups were observed in the ONL at any other PN day studied (from PN17 to PN42).

Finally, the rds retinas showed similar HNE expression to that of the control mice at PN11 in the SL, and it was increased in the rds retinas compared to the control retinas at PN17, PN21, PN28, PN35, and PN42.

#### 3.2.2. Increased Nuclear Oxidative Stress Damage in rds Retina

8-Oxoguanine is a major product of oxidative DNA damage. Interestingly, the structure of 8-oxoguanine is similar to the prevalent ligand for avidin [35]. In our study, avidin was used to try to recognize oxidatively injured DNA because other studies have established that the staining of retinas with 8-oxoguanine antibody and avidin results in the co-labeling of cells in the ONL [35]. Despite this fact, it should be taken into account that avidin also stains cells with a high abundance of biotin and this can interfere with the results obtained.

We observed avidin-positive cells only in the ONL (the layer that contains the nuclei of the cone and rod photoreceptors) of both the control and rds mice. Our results confirm a small number of avidin-positive cells in the control animals, indicating low levels of nuclear oxidative damage. No differences were noticed in the number of avidin-positive cells in the retinas of the control and rds mice at PN11. However, a significant increase in this oxidative stress marker was seen in the rds retina at PN17 compared to the control retinas. The maximum number of avidin-positive cells was observed at PN21 in the rds mice, coinciding with the peak of cell death (8 times higher than the control retinas). At PN28, PN35, and PN42 (when the slower cell death rate period takes place), the number of avidin-positive cells was significantly higher in the rds retinas than in the control retinas only in the midperiphery and near the optic nerve area. No important differences were observed in the number of avidin-positive retinas in the far periphery between the control and rds mice at PN28, PN35, and PN42 (Figure 5A,B).

#### 3.2.3. Glutathione Metabolism Alterations in Rds Retinas

GSH is a tripeptide of glutamate, cysteine, and glycine that exists either in a reduced form or in an oxidized dimeric form (with a disulfide bridge) (GSSG). In this work, we used an antibody that recognizes both reduced and oxidized glutathione.

Immunoreactivity for GSH was noticed in the GCL, IPL, INL, and OPL retinas in both the control and rds retinas (Figure 6B). Interestingly, no immunoreactivity was observed in the ONL and very low expression was observed in the SL (GSH-GSSG expression in these two layers were not quantified because of this reason). The fluorescence intensity analysis for the evaluation of GSH expression was performed using ImageJ software. Immunopositivity for GSH was similar at PN11, PN17, and PN21 in the GCL, IPL, INL, and OPL of the control retinas. At PN28, the degree level of GSH significantly increased in the GCL, INL, and OPL of the control retinas, and then, decreased again at PN35.

The retinas with experimental RP exhibited a similar pattern of labeling to that observed in the control retinas (Figure 6B). However, the observed increase in GSH expression was only observed at PN28 of the GCL of the rds retinas. GSH expression increased earlier (at PN21) in the IPL, INL, and OPL in the rds mice (Figure 6A).

#### 3.2.4. Can the Alterations in Glutathione Be Explained by an Increase in the Synthesis or Transport of Glutathione?

To better understand why GSH and GSSG concentrations were altered in the rds retinas, we evaluated the expression of three different proteins involved in GSH metabolism and regeneration: (i) GCLC; (ii) glutamate–cysteine ligase modulatory subunit; and (iii) 2-oxoglutarate carrier (OGC) (Figure 7).

GSH is synthesized in cells’ cytosol in two steps. The first step is catalysis by the enzyme glutamate–cysteine ligase (GCL) and is the rate-limiting stage in GSH synthesis. This enzyme is made up of two subunits: the catalytic subunit and the modulatory subunit (GCLC and GCLM). The GCLM subunit is enzymatically inactive but plays a critical role in controlling GCL [36]. Although approximately 80–85% of cellular GSH is found in the cytosol, around 10–15% of GSH is present in the mitochondria [37]. One of the most well-studied GSH transporters is 2-oxoglutarate carrier (OGC), which brings GSH from the cytosol into the mitochondria in the substitution of 2-oxoglutarate and other dicarboxylates [38].

GCLC expression in the retinas of the control and rds mice at different postnatal ages can be observed in Figure 7A,B. Our results show that the expression of this enzyme was similar in the control and rds retinas. Nevertheless, a huge increase in the expression of GCLC was found in both groups of mice only at PN17. When GCLM expression was studied (Figure 7A,C), we observed that there were no differences between the concentration of this enzyme in the control and rds retinas at PN11, PN17, PN21, and PN28. However, this concentration was higher at PN28 in both the control and rds mice than at PN11, PN17, and PN21. This retinal concentration remained high at PN35 and PN42 in the control mice, but GCLM expression was significantly lower in the rds group. OGC expression is shown in Figure 7A,D. When the control and rds retinas were compared, no significant differences were observed in OGC expression at any of the ages studied. However, if OGC expression is analyzed only in the rds mice, we can find an increase in the expression of this enzyme at PN35 compared to PN11, PN21, and PN42.

### 3.3. Inflammation

#### 3.3.1. Macroglia Alterations with Disease Progression

RP induces reactive gliosis characterized by the hypertrophy of Müller cells. Typically, gliosis may be described by a dramatic increase in and upregulation of GFAP [37].

The results of GFAP expression in the control and rds groups at the different studied ages can be observed in Figure 8A,B. GFAP was quantified in three different retinal areas (far periphery, midperiphery, and near the nerve). The results from the percentage of area stained with GFAP antibody are given in these three retinal areas, as well as in the total retinal area (Figure 8A). When comparing GFAP expression in the three retinal areas studied and in the total retina, we observed no significant differences between the control and rds mice at PN11 and PN17. However, GFAP expression was significantly higher in the rds retinas at PN21, PN28, PN35, and PN42 than in the controls in all retinal areas. We thus confirm that GFAP increases with disease progression, and that this increase is first detected at PN21, coinciding with the photoreceptor peak of cell death, and after the observed changes in oxidative markers that indicate lipid or nuclear damage.

#### 3.3.2. Changes in Microglia in rds Retinas

Microglia expression was analyzed using the accepted microglia marker Iba1. We analyzed the number of Iba1-positive cells in three retinal areas (far periphery, midperiphery, and near the nerve) and in different layers of the retina (number of Iba1-positive cells/area × 10^6^). Iba1-positive cells were detected in the GCL, IPL, INL, OPL, and ONL.

First, in the GCL layer, a higher number of Iba1-stained cells were observed in the retinas of rds mice at PN11 than in the control mice. The number of these cells decreased, and we could not find any differences between the number of positive cells in the control and rds retinas at PN17. At PN21, PN28, PN35, and PN42, the number of Iba1 cells in the rds retina increased and was significantly higher than the number of Iba1 cells in the control retinas (Figure 9). These results were similar in the three retinal areas studied.

When IPL was studied, we observed no significant differences between the number of Iba1-positive cells in the control and rds retinas at PN11 and PN17. The number of these cells was higher in the retinas of the rds mice than those of the control mice at PN21 and PN28. However, the number of Iba1 cells decreased at PN35 and PN42 in rds mice, and no significant differences were observed with the control mice (Figure 9).

We also quantified the number of Iba1 cells in the INL. The number of these cells was higher in the rds mice than in the control mice in the midperiphery retina at PN11, but not in the far periphery or near the nerve retina. The number of Iba1 cells was also significantly higher in the rds retinas at PN17 and PN21 and was similar to the number of positive cells in the control mice at PN28, PN35, and PN42 (Figure 9) in all three retinal areas studied.

The results regarding this microglia marker observed in the OPL at the beginning of the degeneration process were similar to the results obtained in the GCL. In this sense, a higher number of Iba1-stained cells were found in the retinas of the rds mice at PN11 compared to the control mice. The number of these cells decreased, and we could not find any differences between the number of positive cells in the control retinas and rds retinas at PN17. However, at PN21, PN28, and PN35, the number of Iba1 cells in the rds retina increased and was significantly higher than the number of positive cells found in the control retinas (Figure 9). Interestingly, on the last postnatal day studied (PN42), the number of microglial cells was similar in the rds and control retinas.

In the ONL, the Iba1 cell group was significantly greater in the rds mice than in the control group on all postnatal days, except for PN11 and PN17. The first change and the maximum number of Iba1 cells were observed at PN21 (and later, PN28, PN35, and PN42), and the number of these microglia cells in the rds retinas decreased, but was still significantly greater than in the control retinas (Figure 9).

Finally, Iba1 expression was examined in the total retina (the number of cells found in the entire retinal layer). An increase in the number of microglia cells was identified in the retinas of the rds mice as early as PN11. This increase was also observed on all the other postnatal days studied (PN17, PN21, PN28, PN35, and PN42). The maximum number of microglial cells was observed at PN21, coinciding with the peak of photoreceptor death in these RP mice. It should be noted that the change observed in microglia cells was the first change observed in the rds retinas of all previous oxidative and glial changes studied in this work.

The morphological analysis of the microglia, with the quantification of the ratio of the number of branches/cell and the assessment of the length of the branches, was also performed on the control and rds retinas (Figure 10). The method used for the quantification of branch number and branch length was based on the method developed by Young and Morrison [39].

Regarding the number of branches per soma, if we compare the rds group and the control group, statistically significant differences were only observed from PN21 onwards. In all cases where there was a difference with respect to the controls, a lower number of branches per soma was observed (a significant decline in the number of branches in the rds retinas at PN21, PN28, PN35, and PN42) (Figure 10).

The branch length of the Iba1-positive cells presented a similar distribution to the number of branches. Significantly shorter branches were identified in the rds retinas than in the control group at all ages except PN11 and PN17 (Figure 10).

It is well known that if there is no disease, microglial cells show a ramified morphology with a small, spherical soma, and several branching processes. In response to tissue injury, microglial cells adopt a reactive state distinguished by a shortening of microglial processes and a lack of them (amoeboid form) [40]. Our cellular microglia analysis indicated microglia activation in the rds retinas after PN21 (the peak of photoreceptor death in rds mice). This was supported by the reduced ratio of the number of branches/cell and the length reduction of microglial cell projections.

To understand the role of microglia in RP in the rds animal model, CD68 staining (a marker of activated microglia) was performed (Figure 11). Round cells labeled with the CD68 antibody were observed between the EPR and the ONL (this is easily seen in the magnification photographs, on the left side of Figure 11). These results indicated that microglial cells migrated to the damaged area of the retina. The results showed significantly higher numbers of CD68-positive cells in the rds animals than in the control mice at PN21, PN28, PN35, and PN42 (after the peak of cell death). There was no significant difference in CD68 expression in the control and rds retinas at PN11 and PN17.

## 4. Discussion

Many retinal pathologies, including RP, have been associated with oxidative stress and inflammation. However, a comprehensive analysis of the time-course changes in oxidative stress markers and inflammation and the link between these two processes in rds mice (the animal model that most closely resembles the photoreceptor cell rate observed in human RP) [31] has not yet been performed.

This research aimed to elucidate the time pattern of oxidative stress and microglia distribution and activation in rds mice. The study of ocular oxidative stress and inflammation biomarkers can help us understand the physiopathology of RP and find a good strategy for monitoring the disease. This work can also help us understand when and to what extent oxidative stress and inflammation are part of RP.

In the first part of the present work, we performed a detailed morphometric and immunochemistry analysis to accurately determine the extent of photoreceptor cell death at different time points in the disease progression of rds mice. It is well known that intense mouse-housing light accelerates retinal degeneration in RP mice [34], and therefore, it was necessary to characterize retina cell death in our RP animal model to perform an accurate interpretation of our experimental results. We confirmed that photoreceptor cell death in the retinas of rds mice starts early, as seen in the higher numbers of TUNEL-positive cells in the ONL of the rds retinas compared to the control retinas (Figure 2), and the lower rhodopsin- and annexin-stained cells starting at PN11 (Figure 3). This cell death occurred a few days earlier than the cell death reported by others [30,31], indicating that cell death starts between the second and third postnatal weeks. The results of the hematoxylin–eosin staining, together with those of the TUNEL technique, demonstrate that the peak of photoreceptor death occurs at around PN21, and that there is an accelerated cell death period that lasts until around PN28.

### 4.1. Oxidative Damage to Macromolecules (Lipids and DNA) in the Retinas of rds Mice before the Peak of Photoreceptor Death

We evaluated changes in retinal oxidative stress in the rds mice on the postnatal days selected and compared the results with those of photoreceptor death. Oxidative stress in ocular pathologies usually results from either excessive reactive oxygen and nitrogen species (ROS and RNS) production, altered antioxidant systems, mitochondrial dysfunction, or a combination of these factors [41]. An excess of ROS and/or RNS causes oxidative damage to lipids, DNA, and proteins. To characterize all the factors that accompany oxidative stress, we assessed two markers of oxidative damage to macromolecules (HNE and 8-oxoguanine) and GSH (one of the major intracellular antioxidants) metabolism and mitochondrial transport.

The aldehyde HNE is the peroxidation product of n-6 polyunsaturated fatty acids (PUFAs), such as linoleic and arachidonic acids, and their 15-lipoxygenase metabolites. It has been implicated in various pathophysiological processes and can alter normal cell functions [42]. Therefore, it is interesting to understand the possible alterations in its expression in the retinas of RP animals. Our results showed HNE staining in all retinal layers in both the rds and control mice, with a lower staining intensity observed in the ONL (Figure 4A). We also observed an increase in HNE as early as PN17 (before the peak of cell death and during the accelerated death rate period), and it continued to increase until PN42. This may suggest that oxidative damage to lipids occurs later than the beginning of photoreceptor death but earlier than the peak of retinal death, and that it can be used as a marker to monitor RP.

Oxidative damage to DNA is thought to be of particular importance simply because it can lead to mutations, but the effects of non-mutational oxidation have also been implicated in a wide variety of diseases [43]. The interaction of ROS with DNA induces the formation of different nucleobase modifications. 8-Oxoguanine is a nucleobase-derived lesion that is frequently measured as a marker of oxidative stress [43]. An easy way to measure 8-Oxoguanine is to use avidin to identify oxidatively damaged DNA [35]. This technique has also been validated by other works that confirm that the protein avidin binds preferentially to oxidatively modified DNA [44,45,46].

In this work, we demonstrated that DNA oxidative damage could be observed in the ONL of the rds retinas, and that this damage was detected at PN17 (the same postnatal day on which we detected HNE changes, after the beginning of photoreceptor death but earlier than the peak of retinal death). This increase in avidin-stained cells was temporary, and after PN21, the number of cells marked with avidin decreased (Figure 5A,B).

A previous study used ceruloplasmin and clusterin as markers of oxidative stress [47]. The study investigated changes in these two parameters over time in rds mice, as well as in other RP animal models. They found that oxidative stress markers were elevated to varying degrees throughout the entire period studied (approximately from PN 10 to PN 60). This result is similar to our result obtained with HNE, which increased during the period of time we studied, but the main differences are that it had a different biomarker and that they did not compare their results in rds animals with control ones [47].

Finally, we studied GSH synthesis and transport. GSH is a low-molecular-weight antioxidant that is quickly oxidized to GSSG to detoxify an extensive range of oxidant species. For this purpose, we performed retinal immunohistochemistry using an antibody that recognizes both reduced and oxidized GSH. We found a similar pattern of GSH labeling in the control and rds retinas (Figure 6B); this is compatible with other results reporting the presence of GSH in Müller and horizontal cells [48] and the presence of the trans-sulfuration pathway enzymes cystathionine-γ-lyase and cystathionine β–synthase in these types of cells [49]. Our study showed an increase in GSH-GSSG at PN21 in the IPL, INL, and OPL in the rds mice and at PN28 in the GCL; in all cases, this increase was not permanent and was followed by an immediate decrease (Figure 6A). A previous study performed by Vlachantoni et al. [50] did not find any difference between retinal GSH concentrations in rds mouse retinas compared to control mice at PN30. In that work, the researchers determined reduced GSH spectrophotometrically in retinal homogenates and did not specify where in the retina GSH is expressed. Therefore, comparisons between the study of Vlachantoni et al. and the present study are difficult.

Further studies are needed to know the ratio of GSH to GSSG, but in this work, we tried to focus our attention on the location and metabolism of GSH. For this reason, we investigated GSH synthesis by measuring GCLC and GCLM expression. GCLC was similar in the control and rds retinas on all the postnatal days studied. When GCLM expression was studied, we observed a decrease in the rds mice at PN35 and PN42 (Figure 10). In mammalian cells, three different and well-known mechanisms are used to obtain GSH: (i) de novo synthesis, (ii) extracellular uptake, and (iii) GSSG-reductase catalysis of GSSG [51]. We suggest that the observed increase in GSH-GSSG at PN21 in the rds mice may be a consequence of GSH extracellular uptake or oxidation of GSH to GSSG, rather than a consequence of an increase in its synthesis. Moreover, when the control and rds retinas were compared, no significant differences were found in OGC expression in the retinas of the rds mice, indicating that there was no alteration in the mitochondrial transport of GSH.

Regarding oxidative stress markers, we can conclude that increased markers of oxidative stress to macromolecules are the earliest modified markers. In addition, GSH alterations showed a more complex response and may be considered an adaptive reaction to the oxidative stress triggered at earlier time points.

### 4.2. Microglia Activation after Oxidative Damage to Macromolecules in Retinas of Rds Mice

Gliosis, characterized by the upregulation of GFAP, is a well-studied phenomenon in many different RP experimental models, including the rd1 mouse model [52], the rhodopsin knockout mouse model [53], the rd10 mouse model [19,47], the N-methyl-N-nitrosourea (MNU)-induced model [54], the Royal College of Surgeons (RCS) rats [55], and the rds mouse model [56]. The difference between the various animal models is the extent of this GFAP response. In our case, we observed no significant differences between the control and rds mice at PN11 and PN17. However, GFAP expression was significantly higher in the rds retinas at PN21, PN28, PN35, and PN42 compared to the control retinas (Figure 11). In the rds mice, increases in GFAP have also been observed after PN20 and PN21, respectively, in previous studies [47,57]. These studies are in accordance with the results obtained in this work.

Therefore, the observed macroglia reactivity takes place after the changes in oxidative damage to lipids and DNA, meaning that gliosis is not responsible for, but is a consequence of, photoreceptor death and oxidative stress.

The mean numbers of microglial cells (Iba1-positive cells) were analyzed in the GCL, IPL, INL, OPL, ONL, and entire retinas of the control and rds mice on different postnatal days (Figure 12). An increase in the number of microglia cells was observed in the total retina (and in certain areas of the GCL, INL, and OPL) of the rds mice as early as PN11. This was the first alteration observed in the rds retinas of all the oxidative and macroglial markers studied in this work. This increase was also observed on all the other postnatal days studied (PN17, PN21, PN28, PN35, and PN42) (Figure 12). Though microglia have been studied extensively in other RP animal models, including the rd10 mouse, this is the first study realized that focuses on this characteristic in rds mice.

We investigated the activation/morphology profile changes in microglia during the process of retinal degeneration. We found significantly lower numbers of branches per soma (Figure 10A) and shorter branches (Figure 10B) in the retinal Iba1-positive cells of the rds mice compared to the control animals from PN21 until PN42, and no differences at PN11 and PN17. This may indicate that after PN21, microglia begin to exhibit the typical amoeboid-activated shape with round bodies and scarce dendrites in the retinas of this RP animal model, after photoreceptor death has started and at around the peak of cell death. Similar results were found when studying CD68 staining (a marker of activated microglia) (Figure 11A). We agree with other researchers who have affirmed that the temporal relationship between photoreceptor apoptosis and microglial response suggests that microglia are not responsible for primary photoreceptor death [58]. Nevertheless, microglia activation may be important, as the inhibition of microglia activation by minocycline-reduced photoreceptor apoptosis has been demonstrated in RP experimental animal models [59].

It is easy to understand that oxidative alterations in RP retinas occur before microglia activation, as it has previously been demonstrated that oxidative stress rapidly activates microglia in the central nervous system (CNS) and retinas [60], and that it is one of the first events in the inflammation cascade [61]. However, it is difficult to explain the initial increase in the number of Iba1-positive cells, and a new experiment should be performed to confirm its significance.

## 5. Conclusions

The relevance of this study lies in the fact that we can now sequence the time of oxidative stress and inflammation changes in the rds animal model of RP (Figure 12). This is particularly important as it is known that more than 150 different mutations in Prph2 are known to cause varying forms of rod- and cone-dominant blinding retinal degeneration in patients, and that there is high variability between patients carrying the same mutation [62,63].

The order of oxidative stress and inflammation in rds mice may be summarized as follows: (1) At PN11, there is a small but detectable increase in photoreceptor (rod and cone) death, and a simultaneous increase in microglial cells is observed; (2) later (at PN17), but before the peak of photoreceptor death, damage to macromolecules such as lipids and DNA is observed; (3) at PN21, the maximum photoreceptor death rate is detected along with an increase in GSH-GSSG and GFPA; (4) at PN21, microglial cells are activated; and (5) at PN28, there is a decrease in the expression of GCLM, one of the enzymes responsible for GSH synthesis.

These findings contribute to the understanding of RP physiopathology and highlight oxidative stress and inflammation as potential therapeutic targets for retinal degeneration associated with peripherin mutations. Treatment of RP and other retinal degeneration with antioxidant supplements is controversial, probably because human trials have not clearly demonstrated their efficacy. We must know the extent to which oxidative stress and inflammation are part of degeneration. In this sense, our work suggests that major therapeutic advances could be made if the treatment specifically targets macromolecule oxidative damage rather than GSH depletion or microglia activation. The selection of the antioxidant is crucial. Further studies are necessary, but a good option could be to use superoxide dismutase (SOD) mimetics that would reduce superoxide and have been shown to be protective in other retinal diseases [64,65,66].

## Figures and Tables

**Figure 1 antioxidants-11-01950-f001:**
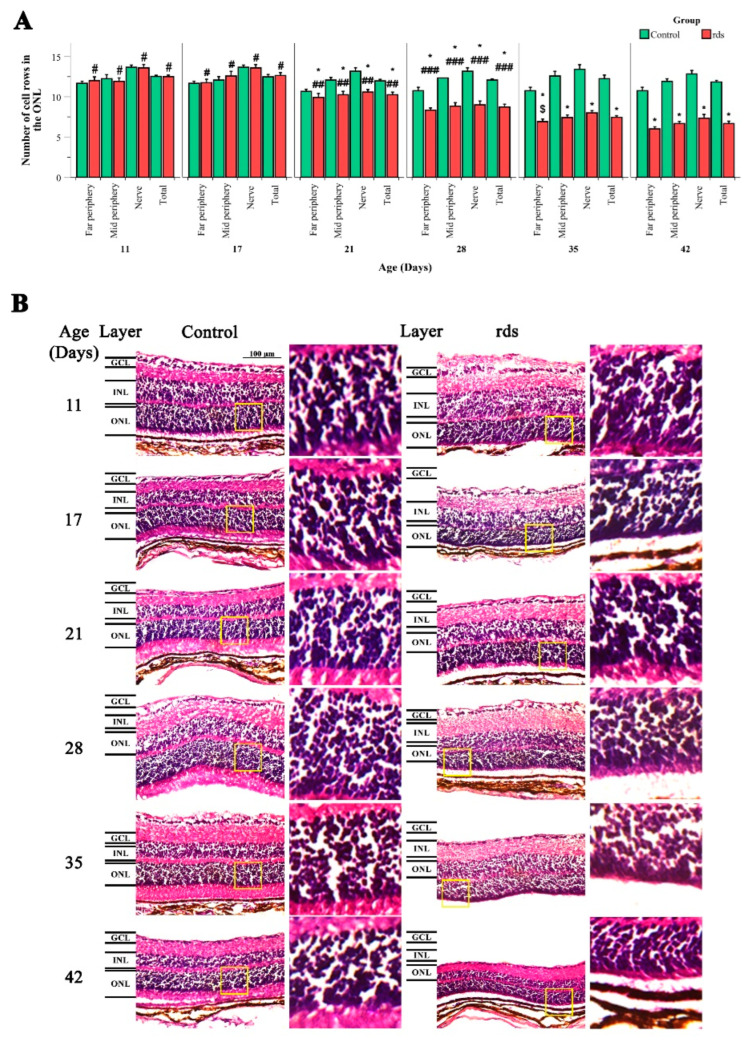
Rows of photoreceptor nuclei in the retinal ONL of control and rds mice. (**A**) Mean number of photoreceptor rows in the control and rds mice; error bars correspond to the standard error of the mean (* *p* < 0.01 vs. control group at the same age; # *p* < 0.0001 vs. rds at PN21, PN28, PN35, and PN42; ## *p* < 0.0001 vs. rds at PN28, PN35, and PN42; ### *p* < 0.05 vs. rds at PN35 and PN42; $ *p* < 0.02 vs. rds at PN42). Four mice per group were used and three histological sections of one retina of each mouse were quantified. (**B**) Retinal hematoxylin–eosin staining of control and rds mice at PN11, PN17, PN21, PN28, PN35, and PN42 in the midperiphery retina.p.

**Figure 2 antioxidants-11-01950-f002:**
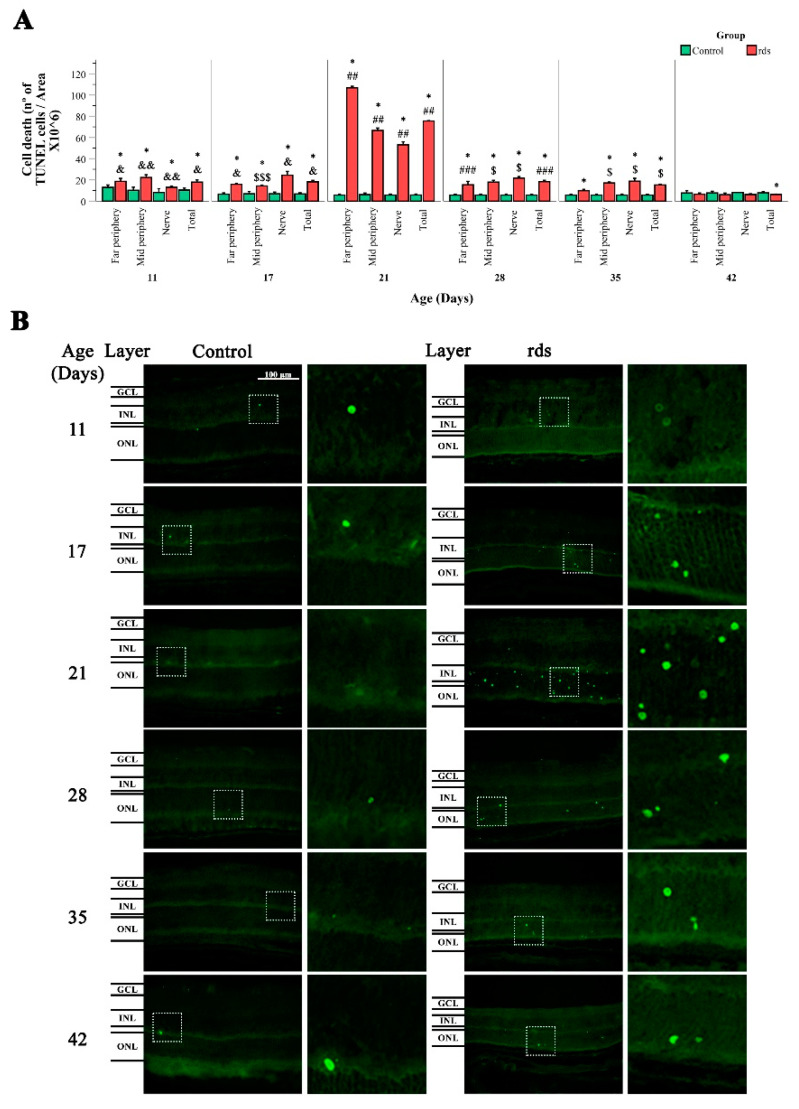
Retinal cell death in control and rds mice. (**A**) TUNEL-positive cells (positive cells × 10^6^/area) at PN11, PN17, PN21, PN28, PN35, and PN42 in the retinas of the control and rds mice in the far periphery, midperiphery, and nerve area. The bars symbolize the means of the number of TUNEL-positive cells, and error bars correspond to the standard error of the mean (* *p* < 0.05 vs. control group at the same age; ## *p* < 0.0001 vs. rds at PN28, PN35, and PN42; ### *p* < 0.01 vs. rds at PN35 and PN42; $ *p* < 0.01 vs. rds at PN42; $$$ *p* < 0.05 vs. rds at PN21 and PN42; & *p* < 0.01 vs. rds at PN21, PN35, and PN42; && *p* < 0.0001 vs. rds at PN21 and PN42). Four mice per group were used and three histological sections of one retina of each mouse was quantified. (**B**) TUNEL staining in retinas from control and rds mice at various postnatal ages (40× magnification photos). Regions shown are midperiphery regions.

**Figure 3 antioxidants-11-01950-f003:**
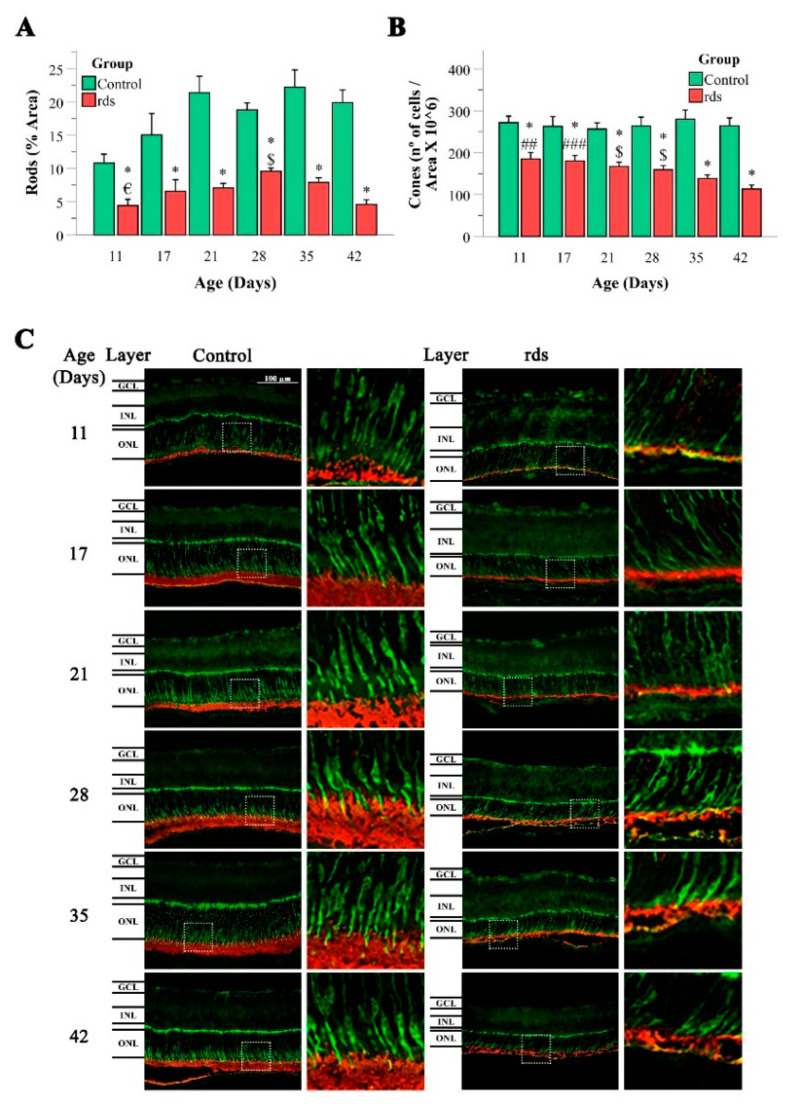
Rhodopsin and cone arrestin staining. (**A**,**B**) Left side: graphical illustration of the area percentage of rhodopsin staining in control and rds retinas, and graphical representation of the number of rods rhodopsin-labeled cells divided by area per 10^6^ (error bars are the standard error of the mean) (* *p* < 0.0001 vs. control group at the same age; € *p* < 0.002 vs. rds mice at PN28; $ *p* < 0.01 vs. rds mice at PN42). Right side: graphical illustration of the number of cells labeled with cone arrestin, divided by area per 10^6^ in the three studied areas and at the different postnatal ages. (* *p* < 0.0001 vs. control group at the same age; ## *p* < 0.001 vs. rds mice at PN28, PN35, and PN42; ### *p* < 0.001 vs. rds mice at PN35 and PN42; $ *p* < 0.01 vs. rds mice at PN42). Four mice per group were used and three histological sections of one retina of each mouse were quantified. (**C**) Microscope 40X magnification photos of rhodopsin and cone arrestin immunostaining in control and rds groups at the different studied ages (rhodopsin in green; arrestin in red).

**Figure 4 antioxidants-11-01950-f004:**
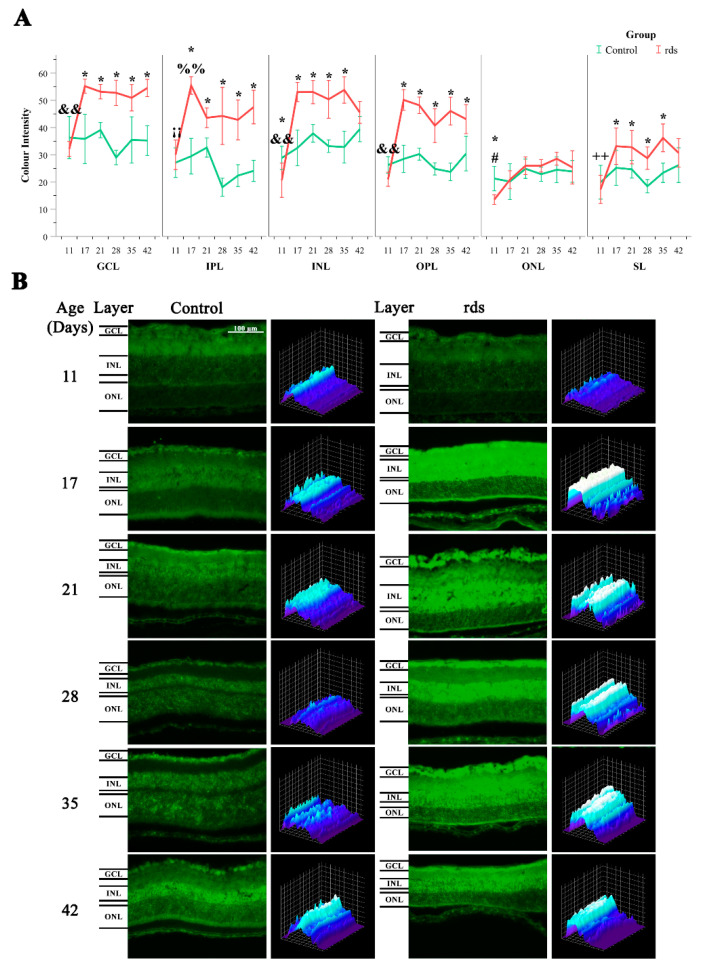
Lipid peroxidation in rds and control retinas. (**A**) Average fluorescence intensity for HNE in the retina in six different retinal layers (GCL, INL, IPL, OPL, ONL, and SL) at PN11, PN17, PN21, PN28, PN35, and PN42. Values are represented as mean density ± SEM for at least 4 mice/group (* *p* < 0.05 vs. control group; # *p* < 0.0001 vs. rds PN21, PN28, PN35, and PN42; && *p* < 0.001 vs. rds PN17, PN21, PN28, PN35, and PN42; %% *p* < 0.05 vs. rds PN35; ++ *p* < 0.05 vs. rds PN17, PN21, PN35, and PN42); ¡¡ p < 0.05 vs. rds PN17, PN21, PN28 and PN42. Four mice per group were used and three histological sections of one retina of each mouse were quantified. (**B**) Representative micrographs of retinal sections and graphical histograms evaluated for lipid peroxidation via immunostaining with HNE antibody at PN11, PN17, PN21, PN28, PN35, and PN42 (40× magnification photos).

**Figure 5 antioxidants-11-01950-f005:**
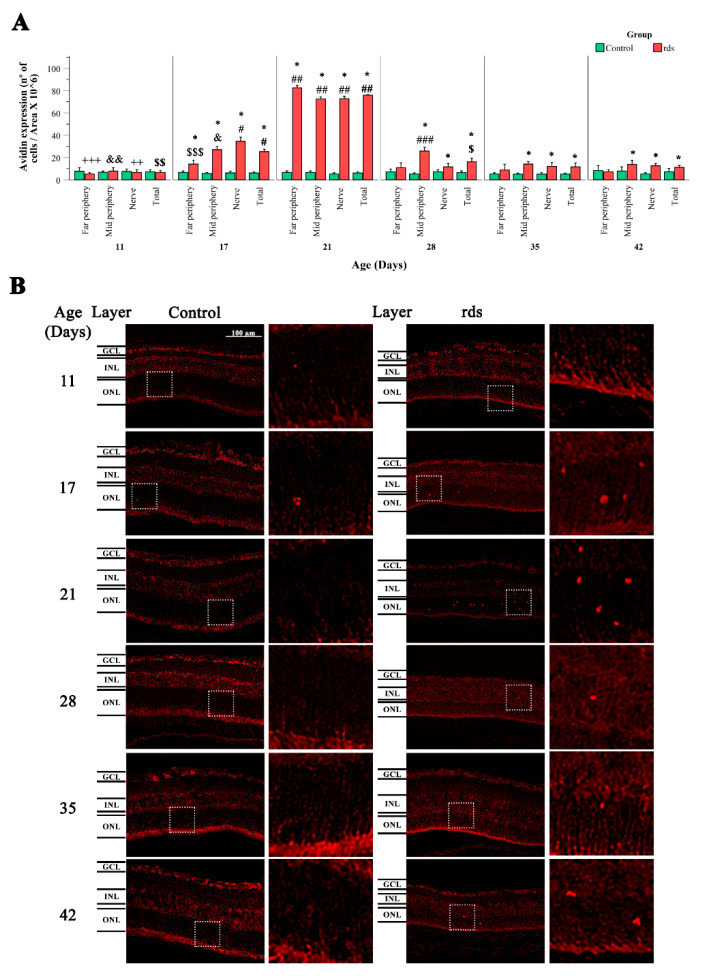
Nuclear oxidative damage in the retina of control and rds retinas. (**A**) Number of retinal avidin-positive cells divided by area per 10^6^ in retinal areas (far periphery, midperiphery, and near the nerve) at different postnatal ages (PN days 11, 17, 21, 28, 35, and 42) (* *p* < 0.05 vs. control group on the same postnatal day; # *p* < 0.0001 vs. rds PN21, rds PN28, rds PN35, and rds PN42; ## *p* < 0.0001 vs. rds PN28, rds PN35. and rds PN42; ### *p* < 0.0001 vs. rds PN35 and rds PN42; $ *p* < 0.05 vs. rds PN42; $$ *p* < 0.0001 vs. rds PN17, PN21, PN28, and PN35; *p* < 0.0001 $$$ vs. rds PN21 and PN42; & *p* < 0.0001 vs. rds PN21, PN35, and PN42; && *p* < 0.005 vs. rds PN17, PN21, PN28, PN35, and PN42; ++ *p* < 0.05 vs. rds PN17, PN21, PN35, and PN42; +++ *p* < 0.0001 vs. rds PN17 and PN21). Four mice per group were used and three histological sections of one retina of each mouse were quantified. (**B**) Representative micrographs of retinal sections evaluated for nuclear oxidative damage using avidin staining in control and rds retinas at PN days 11,1 7, 21, 28, 35, and 42 (microscope 40× magnification photos).

**Figure 6 antioxidants-11-01950-f006:**
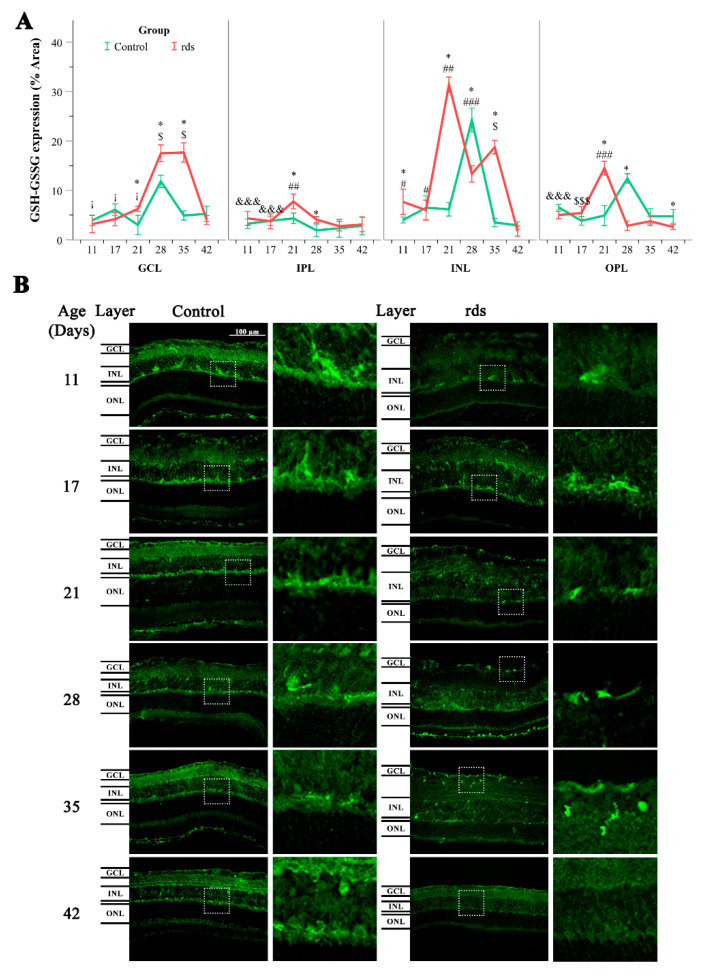
Glutathione expression in the control and rds retinas. (**A**) Glutathione expression (% of stained area) in GCL, IPL, INL, and OPL at different postnatal ages (* *p* < 0.05 vs. control group on the same postnatal day; # *p* < 0.05 vs. rds PN21, PN28, PN35, and PN42; ## *p* < 0.01 vs. rds PN28, PN35, and PN42; ### *p* < 0.0001 vs. rds PN35 and PN42; $ *p* < 0.0001 vs. rds PN42; $$$ *p* < 0.05 vs. rds PN28 and PN35; &&& *p* < 0.05 vs. rds PN21; ¡ *p* < 0.0001 vs. rds PN17). Four mice per group were used and three histological sections of one retina of each mouse were quantified. (**B**) Representative micrographs of retinal sections evaluated for glutathione in control and rds retinas at PN11, PN17, PN21, PN28, PN35, and PN42 (microscope 40× magnification photos).

**Figure 7 antioxidants-11-01950-f007:**
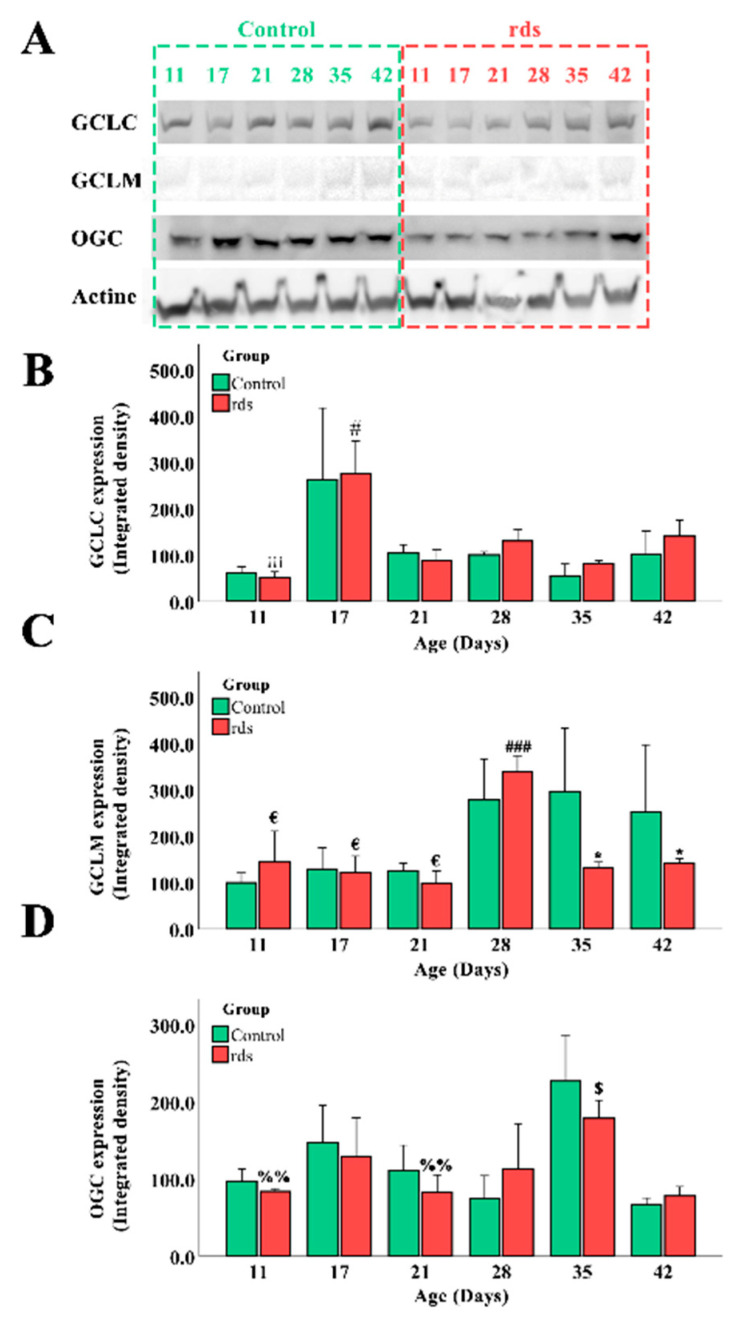
Alteration in proteins involved in GSH metabolism and regeneration. (**A**) GCLC, GCLM, OGC, and Actin expression in retinas of control and rds mice at PN11, PN17, PN21, PN28, PN35, and PN42 (Image Quant™TL photos) (four mice were used per group and both retinas of each mouse were collected and homogenized together). (**B**) GCLC expression in control and rds mice at different postnatal ages (# *p* < 0.0001 vs. rds PN21, PN28, PN35, and PN42; ¡¡¡ *p* < 0.0001 vs. rds PN17). (**C**) GCLM expression in control and rds mice at different postnatal ages (* *p* < 0.05 vs. control group at the same age; € *p* < 0.0001 vs. rds PN28; ### *p* < 0.0001 vs. rds PN35 and PN42). (**D**) OGC expression in control and rds mice at several postnatal ages (%% *p* < 0.0001 vs. rds PN35; $ *p* < 0.0001 vs. rds PN42).

**Figure 8 antioxidants-11-01950-f008:**
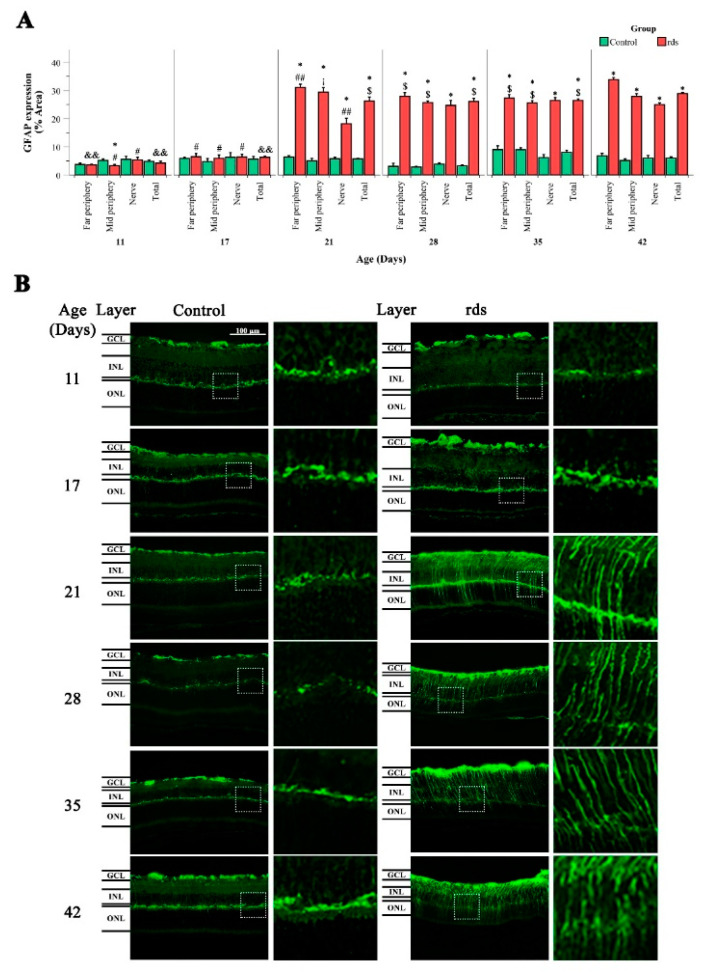
GFAP retinal immunostaining. (**A**) Histogram comparing GFAP (% stained area) in control and rds retinas at PN11, PN17, PN21, PN28, PN35, and PN42 in far periphery, midperiphery, near the nerve, and total retina. Values are represented as mean density ± SEM for at least 4 mice/group (* *p* < 0.0001 vs. control at the same postnatal age; # *p* < 0.005 vs. rds PN21, PN28, PN35, and PN42; ## *p* < 0.005 vs. rds PN28, PN35, and PN42; && *p* < 0.01 vs. rds PN17, PN21, PN28, PN35, and PN42, $ *p* < 0.05 vs. rds PN42; ¡ *p* < 0.0001 vs. rds PN28 and PN35). Four mice per group were used and three histological sections of one retina of each mouse were quantified. (**B**) Images of retinal sections immunostained with GFAP antibody at PN11, PN17, PN21, PN28, PN35, and PN42 (40× magnification) (photographs on the left are control retinas, and photographs on the right are rds retinas).

**Figure 9 antioxidants-11-01950-f009:**
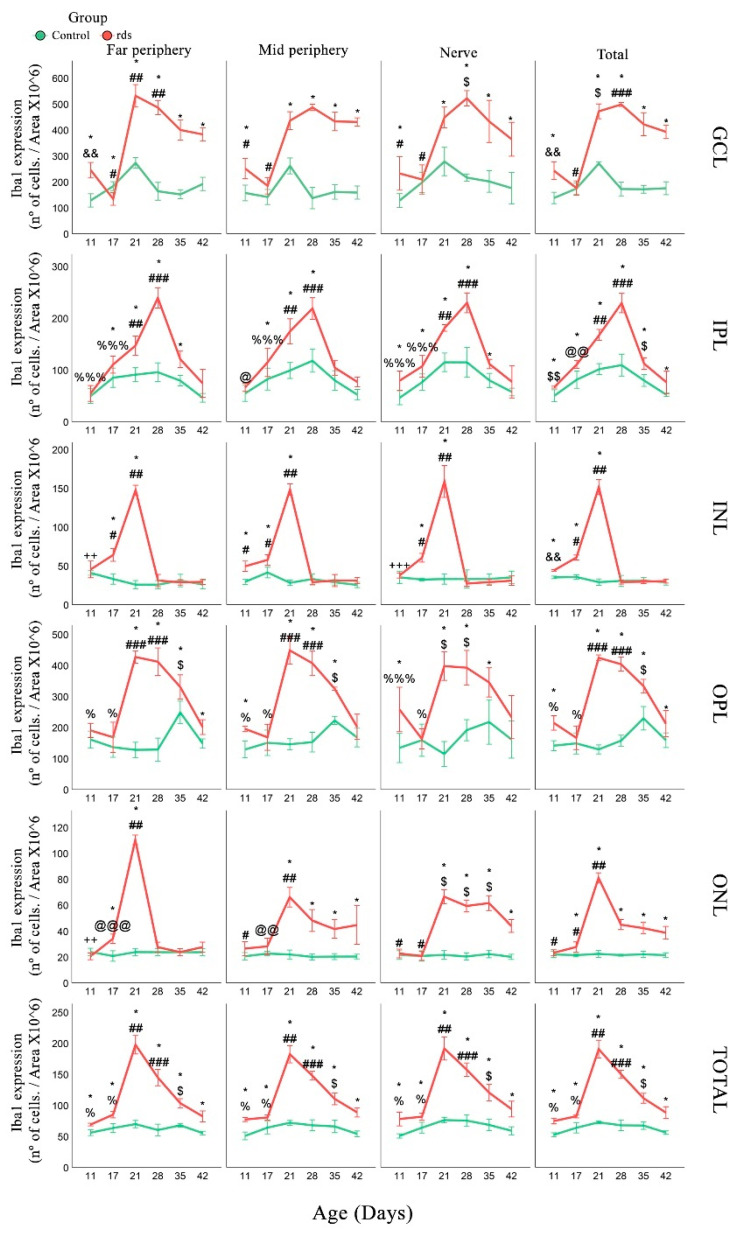
Changes in the number of microglia cells in rds retinas. Number of Iba1-positive cells/areax10^6^ in each retina layer in the three studied areas at PN11, PN17, PN21, PN28, PN35, and PN42 (* *p* < 0.05 vs. control at the same postnatal age; # *p* < 0.05 vs. rds PN21, PN28, PN35, and PN42; ## *p* < 0.05 vs. rds PN28, PN35, and PN42; ### *p* < 0.001 vs. rds PN35 and PN42; $ *p* < 0.05 vs. rds PN42; $$ *p* < 0.001 vs. rds PN17, PN21, PN28, PN35, and PN42; @ *p* < 0.01 vs. rds PN21, PN35, and PN42; @@ *p* < 0.0001 vs. rds PN21 and PN28; @@@ *p* < 0.0001 vs. rds PN21 and PN35; && *p* < 0.0001 vs. rds PN17, PN21, PN28, PN35, and PN42; % *p* < 0.0001 vs. rds PN21, PN28, and PN35; %%% *p* < 0.05 vs. rds PN21 and PN28; ++ *p* < 0.05 vs. rds PN17, PN21, PN35, and PN42; +++ *p* < 0.01 vs. rds PN17 and PN21). Four mice per group were used and three histological sections of one retina of each mouse were quantified.

**Figure 10 antioxidants-11-01950-f010:**
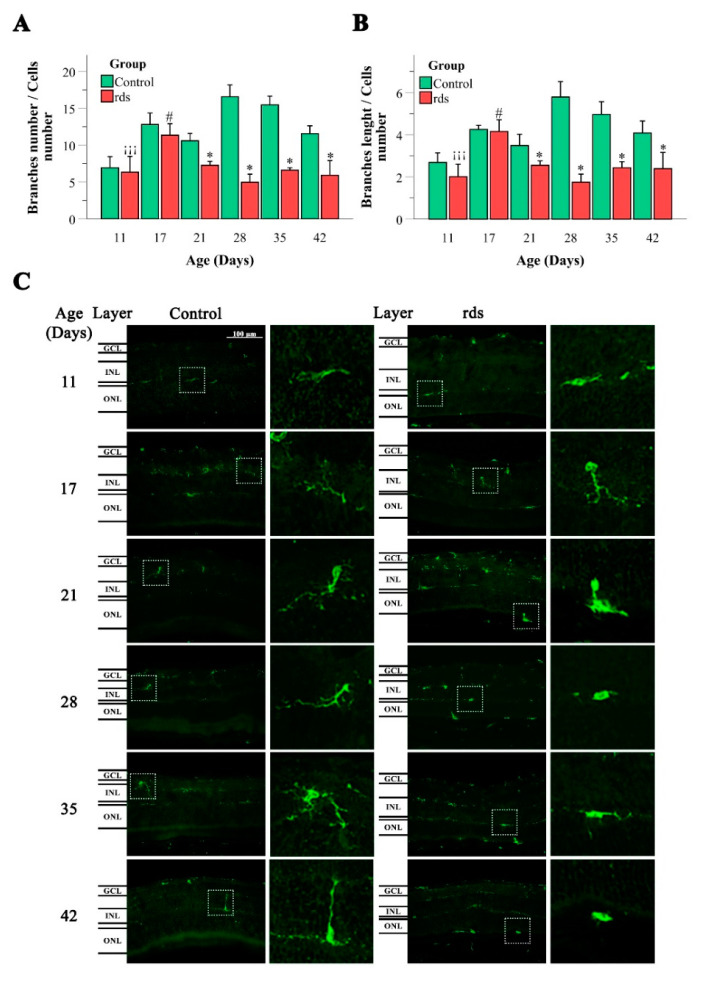
Microglia morphologic analysis. (**A**) Number of branches/Iba1-positive cells in the retinas of control and rds mice at PN11, PN17, PN21, PN28, PN35, and PN42. (* *p* < 0.005 vs. control at the same postnatal age; # *p* < 0.005 vs. rds PN21, PN28, PN35, and PN42; ¡¡¡ *p* < 0.0001 vs. rds PN17). (**B**) Branch lengths of the Iba1-positive cells in the retinas of control and rds mice at PN11, PN17, PN21, PN28, PN35, and PN42. (* *p* < 0.005 vs. control at the same postnatal age; # *p* < 0.005 vs. rds PN21, PN28, PN35, and PN42; ¡¡¡ *p* < 0.0001 vs. rds PN17). Four mice per group were used and three histological sections of one retina of each mouse were quantified. (**C**) Representative micrographs of retinal sections immunostained with Iba1 antibody at PN11, PN17, PN21, PN28, PN35, and PN42 (40× magnification) (photographs on the left side correspond to control mice, and photographs on the right side belong to rds mice).

**Figure 11 antioxidants-11-01950-f011:**
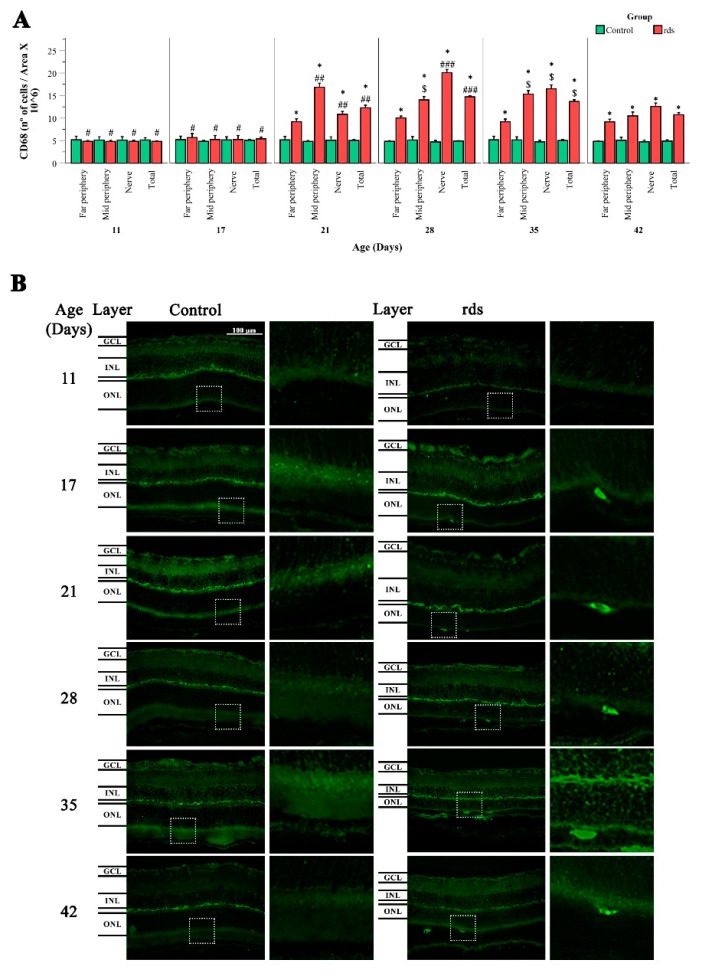
Changes in microglia activation in the retinas of rds mice. (**A**) Number of retinal CD68-positive cells divided by area x 10^6^ in three retinal areas (far periphery, midperiphery, and near the nerve) at different postnatal ages (PN11, PN17, PN21, PN28, PN35, and PN42) (* *p* < 0.0001 vs. control group on the same postnatal day; # *p* < 0.0001 vs. rds PN21, PN28, PN35, and PN42; ## *p* < 0.05 vs. rds PN28, PN35, and PN42; ### *p* < 0.0001 vs. rds PN35 and PN42; $ *p* < 0.0001 vs. rds PN42). Four mice per group were used and three histological sections of one retina of each mouse were quantified. (**B**) Representative micrographs of retinal sections stained with CD68 antibody in control and rds retinas at PN11, PN17, PN21, PN28, PN35, and PN42 (microscope 40× magnification photos).

**Figure 12 antioxidants-11-01950-f012:**
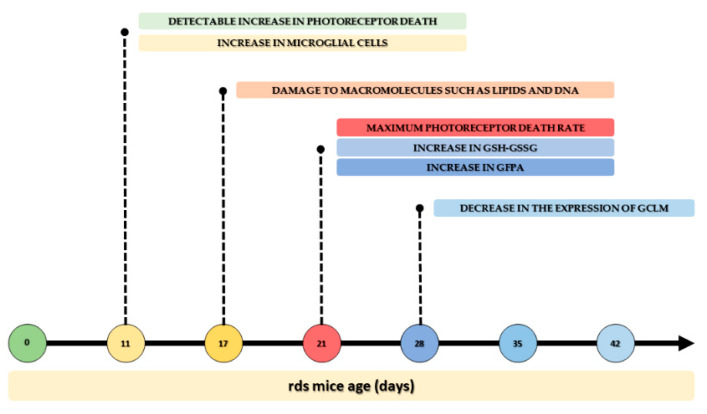
Time sequence of changes in oxidative stress and inflammation in the rds animal model of RP.

**Table 1 antioxidants-11-01950-t001:** Primary and secondary antibodies employed in immunohistochemistry and working concentration.

Antibody	Reference Number	Working Concentration
Alexa Fluor© (594) Conjugated Anti-Rhodopsin 4D1	sc-57432 (Santa Cruz, CA, USA)	1:100
Anti-Cone Arrestin	AB15282 (Merck Millipore, Burlington, MA, USA)	1:500
Rabbit Anti-HNE Antiserum	HNE11-S (Alpha Diagnostic, San Antonio, TX, USA)	1:100
Texas Red Conjugated Avidin	A820 (Molecular Probes Inc., Eugene, OR, USA)	1:200
Rb X Glutathione	AB5010 (Merck Millipore)	1:100
Polyclonal Rabbit Anti-Glial Fibrillar Acidic Protein	Z0334 (Dako, Santa Clara, CA, USA)	1:500
Anti-Iba-1 Rabbit	019-19741 (Wako, Wako, Japan)	1:2000
Rabbit Anti-CD68	ab 125212 (Abcam, Cambridge, UK)	1:500
Alexa Fluor© 488 goat Anti-rabbit IgG	A11008 (Invitrogen, Waltham, MA, USA)	1:200

**Table 2 antioxidants-11-01950-t002:** Primary and secondary antibodies used in Western blot and working concentration.

Antibody	Reference Number	Working Concentration
Rabbit Anti-GCLC	ab 53179 (Abcam)	1:1000
Rabbit Anti-GCLM	ab 124827 (Abcam)	1:1000
Rabbit Anti-SLC25A11	ab 155196 (Abcam)	1:1000
F (ab′) 2–HRP goat Anti-rabbit	31461 (Invitrogen)	1:10,000

## Data Availability

Not applicable.

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
