# Peer review of "Time-Course Changes in Oxidative Stress and Inflammation in the Retinas of rds Mice: A Retinitis Pigmentosa Model"

_antioxidants, 2022, doi:10.3390/antiox11101950_

Round 1

Reviewer 1 Report

This is a well written paper that describes the investigation of the roles of oxidative stress and inflammation in photoreceptor cell death in a mouse model of retinitis pigmentosa, specifically the rds mouse. The authors perform a number of relevant assays including investigation of lipid peroxidation, DNA oxidative damage, GSH/GSSG, GSH synthesis enzymes, GFAP and IBA1, and CD68. The authors investigate photoreceptor cell death at various post-natal days in the rds mouse and correlate photoreceptor cell death with oxidative stress and inflammation. The data presented support the conclusions. I have the following suggestions:

1. The authors should consider a figure that shows the timeline of events as outlined in the conclusions. This figure could show the relationship of oxidative stress and inflammation with each other and with photoreceptor death.

2. The authors state in the last sentence of the abstract that "These findings... help to understand whether oxidative stress and inflammation are therapeutic targets". The abstract and the article itself should end with a more definitive statement regarding whether or not these are therapeutic targets.

3. The authors should consider performing experiments to determine whether oxidative stress (separate from inflammation) can be mitigated and result in reduced photoreceptor cell death.  Such experiments could be conducted using SOD mimetics, which would reduce superoxide and perhaps be protective.

Author Response

We appreciate the comments made by the reviewers in our manuscript “Time-course changes in oxidative stress and inflammation in the retinas of rds mice, a retinitis pigmentosa model” and we thank them for their time.

We think that these suggestions improve considerably the quality and clarity of our manuscript. The manuscript has been thoroughly analyzed and comments for reviewers detailing the changes have been made. We have also highlighted the changes made in the manuscript in red. We hope that that our answers to these comments match the level of the questions. 

Reviewer 1

Comments and Suggestions for Authors

This is a well-written paper that describes the investigation of the roles of oxidative stress and inflammation in photoreceptor cell death in a mouse model of retinitis pigmentosa, specifically the rds mouse. The authors perform a number of relevant assays including investigation of lipid peroxidation, DNA oxidative damage, GSH/GSSG, GSH synthesis enzymes, GFAP and IBA1, and CD68. The authors investigate photoreceptor cell death at various post-natal days in the rds mouse and correlate photoreceptor cell death with oxidative stress and inflammation. The data presented support the conclusions.

I have the following suggestions:

  1. The authors should consider a figure that shows the timeline of events as outlined in the conclusions. This figure could show the relationship between oxidative stress and inflammation with each other and with photoreceptor death.

We agree with the reviewer and have added a new figure to the manuscript (Figure 12, line 758).

The figure is the one that we show below:

Figure 12. Time-sequence of changes in oxidative stress and inflammation in the rds animal model of RP.

  1. The authors state in the last sentence of the abstract that "These findings... help to understand whether oxidative stress and inflammation are therapeutic targets". The abstract and the article itself should end with a more definitive statement regarding whether or not these are therapeutic targets.

We agree with the reviewer, so the final sentence of the abstract has been changed to “These findings contribute to understand that, in RP, oxidative stress and inflammation evolution and relation are dynamic time depending on and highlight both processes as potential therapeutic targets in this disease.” and can be seen in lines 23 to 25. In the same sense, the first sentence of the last paragraph in conclusion has been changed to “These findings contribute to the understanding of RP physiopathology and highlight oxidative stress and inflammation as potential therapeutic targets for retinal degeneration associated with peripherin mutations.” and it can now be read in line 765.

  1. The authors should consider performing experiments to determine whether oxidative stress (separate from inflammation) can be mitigated and result in reduced photoreceptor cell death. Such experiments could be conducted using SOD mimetics, which would reduce superoxide and perhaps be protective.

We absolutely agree with the reviewer, considering that these dates are part of a bigger research project which will continue with further papers we will take heavily this advice and it will be considered to be included as a future experiment. Anyway, we have added some considerations regarding this subject and now in lines 772 to 775 the following sentence can be read: “. The selection of the antioxidant is crucial. Further studies are necessary, but a good option could be to use superoxide dismutase (SOD) mimetics that would reduce superoxide and have been shown to be protective in other retinal diseases [65-67].”

Reviewer 2 Report

The manuscript has a potential to be important contribution to characterisation of temporal changes in oxidative stress, gliosis, microglia/macrophage activation and photoreceptor cell loss in rds mouse. As specified below, there are some issues which needs to be addressed.

1.      Materials and Methods: the genetic background of the control and rds mice needs to be stated

2.      As rightly pointed by the authors, the lighting conditions can affect the retinal degeneration. Irradiance spectra of light the mice were exposed to or at least the total irradiance and the type of light used (fluorescent or LED based with colour temperature given on each bulb).

3.      It is not stated how animals were killed, how quickly the eyes were immersed in PFA and whether any steps were undertaken to facilitate PFA penetration into the eye.

4.      It is not clear why PN11, PN17, PN21, PN28, PN32, and PN42 were selected for evaluation of retinal changes. It appears from the description of the time-course of changes occurring in the rds retina that extending the investigation to 12 months and including time points at 3, 6 and 9 months would be very informative in correlating the oxidative stress, inflammation and cell loss.

5.      Line 158-159: the localisation of the areas where the row of nuclei and (immune)fluorescence were quantified needs to be more precise. How far from the ora serrata and over what distance were the far- and mid-periphery quantified? How far from the edge of the optic disc and over what length were the “nerve” areas quantified?

6.      Lines 182-186: The following sentences do not make sense: “To evaluate changes in rod cell numbers, reduced and oxidized glutathione (GSH- GSSG) expression, and macroglia expression (GFAP), the percentage of area occupied by these proteins was determined. To quantify cone cell numbers, avidin-positive cells, microglial changes (Iba1), and macrophage presence (CD68), positive cells per area unit were 185 studied.” How the total free glutathione and GFAP would reflect the changes in rod numbers, and avidin-positive cells, microglial changes (Iba1), and macrophage presence (CD68) serve for quantification of cones?

7.      Lines 182-184 (the first sentence above). Were fluorescence intensities ignored for quantification of total free glutathione and GFAP expression?

8.      Line 188-189: This sentence needs clarification: “Rod cells, cone cells, HNE, and GSH-GSSG were quantified only in the nerve section because of its regular expression.” No rods and cones are expected to be present in the nerve section.  The term “nerve section” needs to be replaced by “section close to the optic disc” or “central retina section.” It is also not clear what is meant by “regular expression.”

9.      Lines 190-191: “GSH-GSSG, Iba1, and HNE were quantified in each retina layer individually.” From the Results it appears that the quantification was done only in 5 layers: GCL, IPL, INL, OPL, ONL, while the other four layers: NFL, ISL, OSL and RPE were not analysed, so state here exactly which layers were analysed.

10.  The description of fluorescence quantification needs to include more detail. What areas were analysed in Image J? The whole layer of interest in the entire section or smaller areas? Was the person imaging and quantifying fluorescence blind to which group of animals the sections belonged to?

11.  Line 193-194 It stated that both retinas of each mouse were used for the Western blotting. It needs to be stated how many mice from each experimental group were used. It is stated earlier, in line 137-138, that at least 4 animals were used in each group. It would mean that if three mice from the same group of four were used for Western blotting, only one animal was left for histology and immunofluorescence.

12.  Line 222 and further in the text and figures: PN35 is listed here whereas in the Material and Methods there is no mention of evaluation of the retina at this time point.

13.   Figures 1, 2, 3, 5, 6, 9, 10 and 11 or their captions (or Methods section) need to include information on how many mice from each group, how many eyes, and how many sections from each eye were used.

14.  Fig. 3 – the resolution of the images is no high enough to distinguish individual rods so it is not clear how their numbers were quantified.

15.  It needs to be explained why the assessment of lipid peroxidation and glutathione was not done for the inner and outer segment layers.

16.  Fig. 4, its caption and description of the HNE quantification in the Methods are contradicting each other. The label of the vertical axis: “Colour intensity (Green pixels %)” What was quantified: the average fluorescence intensity over the selected area or number of pixels with fluorescence intensity above a set threshold?

17.  Fig. 4, 8 or their captions (or Methods section) state that at least 4 mice from each group were used; additional information needed is how many eyes, and how many sections from each eye were used.

18.  Fig. 5. It is not clear why avidin was used for detection of 8-hydroxy deoxyguanosine and not its antibody. As expected, avidin stains every cell due to high abundance of biotin. The Sanz et al. paper cited by the authors,  also shows the staining with biotin of all retinal cells and only some cells showing co-localisation of that staining with antibody-based staining of 8-OHdG.

19.  Evaluation of glutathione would benefit from evaluation of the ratio of GSH to oxidized glutathione or total glutathione.

20.  Fig. 7 suggests that the same nitrocellulose membrane was used for all three primary antibodies. It is not described this way in the Methods section.

21.  Fig. 7 caption needs to include information how many mice from each group were used.

22.  Fig. 10 and methods description: It is not clear how the number of microglia branches and branch length could be quantified in retinal sections. Retinal flatmounts and collecting z-stack of images would enable such a task. 

23.  Fig. 11. CD68 immunostaining appears highly non-specific with the greatest fluorescence in the OPL.

24.  Controls images without primary antibodies need to be included and, in case they show some fluorescence also controls without 2ndary antibodies to see if it is due to oxidized lipids or lipofuscin. 

25.  The Discussion is largely a summary of the Results. It would benefit from comparing the findings with what has been reported so far in the literature regarding rds mouse as well as discussing their physiological relevance for the peripherin-associated degeneration of the human retina.

26.  What is meant to be the original images of Western blots is the same image as the Fig. 7 included in the manuscript, just with cropped left-hand side. There are no images from other mice experiments based on which the statistical differences were calculated.

27.  Supplementary tables include a lot of symbols with no explanation what they stand for. Do they provide any more information than that provided already in the figures and their captions?

28.  There are some phrases with do not make sense:

Abstract: “macromolecules oxidative damage”

Starting in line 132-134: “The control and care of the micesaccomplished the Association for Research in Vision and Ophthalmology (ARVO) Statement for the Use of Animals in Ophthalmic and Vision Research.”

Line 530-531: “inflammation ocular biomarkers”

29.  There are some spelling and other typographical errors. For example in lines 160, 161: “Deoxinucleotidyl” instead of “deoxynucleotidyl”

Author Response

Reviewer 2

Comments and Suggestions for Authors

The manuscript has the potential to be an important contribution to the characterisation of temporal changes in oxidative stress, gliosis, microglia/macrophage activation and photoreceptor cell loss in rds mice. As specified below, there are some issues which need to be addressed.

We appreciate the comment made by the reviewer and the suggestions in the manuscript “Time-course changes in oxidative stress and inflammation in the retinas of rds mice, a retinitis pigmentosa model”. All the requests of reviewer 2 have been considered in order to improve our paper.

  1. Materials and Methods: the genetic background of the control and rds mice needs to be stated.

As suggested by reviewer 2 we have included a part in the material and method section which include the genetic background of control and rds mice.  Now in line 131 to 133 it can be read as follows: “Control (C3Sn.BLiA-Pde6b<+>/DnJ, homozygous for Pde6b<+>) and rds mice (C3A.CG-Pde6b<+> Prph2<Rd2>/J, homozygous for Prph2<Rd2>) were used in this work. Mice derived from the Jackson Laboratory colony (The Jackson Labs, Bar Harbor, ME, USA).”

  1. As rightly pointed out by the authors, lighting conditions can affect retinal degeneration. Irradiance spectra of light the mice were exposed to or at least the total irradiance and the type of light used (fluorescent or LED based with colour temperature given on each bulb).

Our animal facility is provided by led bulbs which have an irradiance between 20 and 30 lux. We have included this information in the material and methods section. Now (line 138-140) it can be read: “Light illuminance was provided by led bulbs and determined at the cage level with a dig-ital lux meter (Digital Light Meter, Dr.Meter, Santa Cruz CA 95060 USA). Light intensity was found to be between 20 and 30 lux.”

  1. It is not stated how animals were killed, how quickly the eyes were immersed in PFA and whether any steps were undertaken to facilitate PFA penetration into the eye.

We absolutely agree with reviewer 2. We have added all the details those details in line 157 to 159: “Mice were euthanized by neck dislocation. The eyes were enucleated, the cornea of each eye was pierced with a needle and then, the eye was immediately embedded in paraformaldehyde (4% PFA) (4% PFA) during 2 h.”

  1. It is not clear why PN11, PN17, PN21, PN28, PN32, and PN42 were selected for evaluation of retinal changes. It appears from the description of the time-course of changes occurring in the rds retina that extending the investigation to 12 months and including time points at 3, 6 and 9 months would be very informative in correlating the oxidative stress, inflammation and cell loss.

The explanation of why we have chosen these ages is because these days are important time points in normal retinal development. Information has been added to the material and methods section of the manuscript in line 148 to 155: “These postnatal days have been selected because: 1) it has been reported that initial changes in macrophages are visible in the inner retina at PN11 in rds retinas;  2) the photoreceptor peak of cell death in rds retina occurs around PN21; 3) PN17 was considered interesting because is a intermediate state between first changes and peak of cell death; 4) in rds retinas and that there is an accelerated cell death period that lasts until PN28: 5) A slower cell death rate period in rds photoreceptors starts around PN35 and 6) PN42 was selected to monitor what happens during the slow photoreceptor death period [30, 31].”

Our experiment has not been extended to further ages because the initial purpose of our research is to find the relationship between cellular death, oxidative stress and inflammation in the early stage of retinitis pigmentosa in rds mice. However, we will take into account the recommendation of the reviewer for further research projects.

  1. Line 158-159: the localisation of the areas where the row of nuclei and (immune)fluorescence were quantified needs to be more precise. How far from the ora serrata and over what distance were the far- and mid-periphery quantified? How far from the edge of the optic disc and over what length were the “nerve” areas quantified?

The location of the studied areas in the retina is the far periphery, which is beside the ora serrata; the nerve area, which is beside the optic nerve; and the mid periphery, which is in a midpoint between both mentioned areas. The distance occupied by each area is around 800 micrometers. To clarify this subject, we have added one supplementary figure (Supplementary Figure 1) to the manuscript.

  1. Lines 182-186: The following sentences do not make sense: “To evaluate changes in rod cell numbers, reduced and oxidized glutathione (GSH- GSSG) expression, and macroglia expression (GFAP), the percentage of area occupied by these proteins was determined. To quantify cone cell numbers, avidin-positive cells, microglial changes (Iba1), and macrophage presence (CD68), positive cells per area unit were 185 studied.” How the total free glutathione and GFAP would reflect the changes in rod numbers, and avidin-positive cells, microglial changes (Iba1), and macrophage presence (CD68) serve for quantification of cones?

We agree with the reviewer that the sentence can lead the reader to a misunderstanding, and we have changed the original sentence: “Changes in rod cell numbers, reduced and oxidized glutathione (GSH-GSSG) expression and macroglia expression (GFAP), where evaluated by the percentage of area occupied by these proteins. Cone cell numbers, avidin-positive cells, microglial changes (Iba1), and macrophage presence (CD68), were studied by counting the number of positive cells per area unit.” (Lines 203 to 207).

  1. Lines 182-184 (the first sentence above). Were fluorescence intensities ignored for quantification of total free glutathione and GFAP expression?

GFAP and glutathione were determined measuring the area occupied by the fluorescence. In this case, the taken photos were always performed in the same conditions, including intensity. This information has been added to the manuscript (line 232).

  1. Line 188-189: This sentence needs clarification: “Rod cells, cone cells, HNE, and GSH-GSSG were quantified only in the nerve section because of its regular expression.” No rods and cones are expected to be present in the nerve section. The term “nerve section” needs to be replaced by “section close to the optic disc” or “central retina section.” It is also not clear what is meant by “regular expression.”

We agree with the reviewer, so he has changed those expressions by the sentence: “Rod cells, cone cells, HNE, and GSH-GSSG were quantified only in the section close to the optic nerve because of its expression is similar in all the retinal sections. “(lines 212 and 213).

  1. Lines 190-191: “GSH-GSSG, Iba1, and HNE were quantified in each retina layer individually.” From the Results it appears that the quantification was done only in 5 layers: GCL, IPL, INL, OPL, ONL, while the other four layers: NFL, ISL, OSL and RPE were not analysed, so state here exactly which layers were analysed.

We agree with the reviewer, and we have stated the layers in which we have performed the quantification. Now in lines 215 to 219, it can be read as follows:

“HNE was quantified in the in six different layers of the retina: ganglion cell layer (GCL), inner plexiform layer (IPL), inner nuclear layer (INL), outer plexiform layer (OPL), ONL and segment layer. GSH was quantified in the GCL, IPL, INL and OPL (no or very low expression was found in the ONL and SL) and finally Iba1 was quantified in the GCL, IPL, INL, OPL and ONL.”

  1. The description of fluorescence quantification needs to include more detail. What areas were analysed in Image J? The whole layer of interest in the entire section or smaller areas? Was the person imaging and quantifying fluorescence blind to which group of animals the sections belonged to?

As reviewer 2 suggested we have added a section in which we describe fluorescence quantification with Image J with more detail:

“Photos which have been taken at 20X magnification were analyzed using the software ImageJ Fiji version 1.52p.

Number of TUNEL, avidin, CD68 positive cells were counted per unit of area, and the entire retina was used for these quantifications. Iba1 positive cells were counted per unit of area in five different retinal layers (GCL, IPL, INL, OPL and ONL). Iba1 branches number and length were analyzed using the above-mentioned software ImageJ which is able to take an image and draw a line skeleton inside the image and allow us to measure the number of branches per cell body and the length of these branches.

GSH-GSSG was estimated determining the occupied percentage area of the staining in four retinal layers (GCL, IPL, INL, OPL and ONL), but previously a threshold was established. GFAP was measured similarly to GSH-GSSG and occupied percentage area of the staining was estimated in the entire retina. Finally, the quantification of HNE was performed by studying the intensity of the staining in the following retinal layers: GCL, IPL, INL, OPL, ONL and SL.”

All this information can now be read in lines 220 to 234.

  1. Line 193-194 It stated that both retinas of each mouse were used for the Western blotting. It needs to be stated how many mice from each experimental group were used. It is stated earlier, in line 137-138, that at least 4 animals were used in each group. It would mean that if three mice from the same group of four were used for Western blotting, only one animal was left for histology and immunofluorescence.

We apologize for the misunderstanding. In our study we have used four mice in each group for histological and immunohistochemistry studies and another four mice per group were used for western blot analysis. We have changed the two sentences mentioned by the reviewer in the new version of the manuscript to clarify this question (lines 147-148 and 237).

  1. Line 222 and further in the text and figures: PN35 is listed here whereas in the Material and Methods there is no mention of evaluation of the retina at this time point.

We agree with the reviewer. There was a mistake in the materials and methods section in which it was described as a studied age PN32 instead of PN35. The mistake has been solved in the new version of the manuscript.

  1. Figures 1, 2, 3, 5, 6, 9, 10 and 11 or their captions (or Methods section) need to include information on how many mice from each group, how many eyes, and how many sections from each eye were used.

We agree with reviewer 2 and we have included this information in all the mentioned figure legends. 4 mice per group were used. 3 histological sections were selected from one eye. These sections were quantified and the mean from this quantification was taken as the value of the mouse. In the graphics what we see is the is mean from the 4 mice used in the experiment and the standard deviation.

  1. Fig. 3 – the resolution of the images is not high enough to distinguish individual rods so it is not clear how their numbers were quantified.

Due to the addition of the image in a word document the quality of the image has substantially decreased. But it is important to notice that the number of cone photoreceptors could be counted paying attention to the pedicles in the basement of the cone and that the nucleus is stained with a lighter green compared to the cell body.

This information has been explained now in the material and methods section in line 209 to 211-

  1. It needs to be explained why the assessment of lipid peroxidation and glutathione was not done for the inner and outer segment layers.

Due to the mutation of our mice model which is the protein peripherin 2 present in the inner and outer segments of the photoreceptors our animal model is not able to develop these subcellular structures or at least the mice can’t develop it in the same way as in control mice making the measurement and the comparison extremely difficult. However, we agree with the reviewer, and we have reanalyzed the images. We have quantified HNE expression in the segment layer (though we have not been able to quantify independently inner and outer segments). Our results demonstrate an increase in HNE expression in rds retinas at PN17, 21, 28, 35 and 42 and we have written these results in the new version of the manuscript (lines 361 to 363). Regarding GSH, we have not found any expression t the ONL and the expression at the SL was too low to be quantified (lines 410 to 412).

  1. Fig. 4, its caption and description of the HNE quantification in the Methods are contradicting each other. The label of the vertical axis: “Colour intensity (Green pixels %)” What was quantified: the average fluorescence intensity over the selected area or number of pixels with fluorescence intensity above a set threshold?

We agree with reviewer 2. The quantification used in HNE quantification was intensity. In this sense, we have corrected the figure in order to make it clearer.

  1. Fig. 4, 8 or their captions (or Methods section) state that at least 4 mice from each group were used; additional information needed is how many eyes, and how many sections from each eye were used.

We agree with reviewer 2 and we have added additional information to the figure text in order to clarify this question: 4 mice per group were used. 3 histological sections were selected from one eye. These sections were quantified and the mean from this quantification was taken as the value of the mouse. In the graphics what we see is the is mean from the 4 mice used in the experiment and the standard deviation.

  1. Fig. 5. It is not clear why avidin was used for detection of 8-hydroxy deoxyguanosine and not its antibody. As expected, avidin stains every cell due to high abundance of biotin. The Sanz et al. paper cited by the authors, also shows the staining with biotin of all retinal cells and only some cells showing co-localisation of that staining with antibody-based staining of 8-OHdG.

We agree with reviewer 2. However, the work by Sanz et al. explains that using avidin staining is a method for 8-OHdG detection. We have also added other references of publications that confirm the validity of this method (line 656). We have also used directly 8-OHdG, but, in our experience (as in Sanz et al. work) this method needs a treatment with proteinase K before the 8-OHdG staining with the purpose to create pores in the cell membranes and allow the antibody to go inside the cell. This previous treatment with proteinase K has led us to tissue destruction which in the end results in irregular results and quantification problems. We also consider, that the problems mentioned by the reviewer do not invalidate our results, as the main objective of our work is to detect when changes start and not total quantification.

  1. Evaluation of glutathione would benefit from evaluation of the ratio of GSH to oxidized glutathione or total glutathione.

We absolutely agree with reviewer 2. In our article, we try to focus our attention on the location and metabolism of the GSH. It will be interesting to know the ratio of GSH-GSSG and we will consider measuring this value in further research projects and we have added this consideration in the discussion section of the manuscript (lines 684 and 685).

  1. Fig. 7 suggests that the same nitrocellulose membrane was used for all three primary antibodies. It is not described this way in the Methods section.

This membrane was performed to make a single image for all the antibodies and studied groups and simplify the understanding of the results. All the original membranes have been now uploaded.

  1. Fig. 7 caption needs to include information on how many mice from each group were used.

In the western blot analysis, 4 mice were used per group. Both retinas of each mouse were collected and homogenized together. This information has been added to figure 7 caption.

  1. Fig. 10 and methods description: It is not clear how the number of microglia branches and branch length could be quantified in retinal sections. Retinal flatmounts and collecting z-stack of images would enable such a task.

The method used for the quantification of branches number and branches length was based on the method developed in the paper “Young, K.; Morrison, H. Quantifying Microglia Morphology from Photomicrographs of Immunohistochemistry Prepared Tissue Using ImageJ. J. Vis. Exp. 2018, 2018, doi:10.3791/57648.”. This method is based on an application of the software ImageJ which is able to take an image and draw a line skeleton inside the image taking in consideration the shape of the cell. Within this line skeleton some parameters are quantified automatically, and we have selected the number of branches per cell body and the length of these branches. We have added some of this information to the new version of the manuscript (lines 224 to 227).

We agree with reviewer 2 that if we would have used the flat mount method maybe we will have a better perspective of the Iba1 cells’ ramifications, but it is important to notice that we also wanted to know the migration of Iba1 cells to the damaged part of the retina which in our case is the ONL. Finally, in our case, even though we have used z-stacks to collect the images we assume that we are using one dimension.

  1. Fig. 11. CD68 immunostaining appears highly non-specific with the greatest fluorescence in the OPL.

As reviewer 2 has noticed the staining of the CD68 marker may be seen as non-specific. However, it could be observed easily that there are some round cells between the EPR and the ONL, indeed all the magnification squares in the figures are in this position. These results indicate (as reported by other authors) that have microglial cells have migrated to the damaged area of the retina. We have written this information in the new version of the manuscript (lines 585 to 588).

  1. Controls images without primary antibodies need to be included and, in case they show some fluorescence also controls without 2ndary antibodies to see if it is due to oxidized lipids or lipofuscin.

As the reviewer suggested in our staining, we always add negative controls with the secondary antibody. Most of the immunohistochemistry techniques have been performed using the same secondary antibody (Alexa Fluor© 488 goat An-ti-rabbit IgG). We have made a new supplementary figure with an image corresponding to a negative control of a PN17 rds retina. The other secondary antibody was used only for rods quantification and was Alexa Fluor© (594) Conjugated Anti-Rhodopsin 4D1 and we have not performed a negative control because it is a conjugated antibody.

  1. The Discussion is largely a summary of the Results. It would benefit from comparing the findings with what has been reported so far in the literature regarding rds mouse as well as discussing their physiological relevance for the peripherin-associated degeneration of the human retina.

We have changed the discussion part and included other studies performed in rds mice and compared their results with others. We have also highlighted the relevance of the peripherin-associated degeneration of the human retina (lines 745 to 739).

  1. What is meant to be the original images of Western blots is the same image as the Fig. 7 included in the manuscript, just with cropped left-hand side. There are no images from other mice experiments based on which the statistical differences were calculated.

We provide the reviewer with the original images that were used for the calculation.

  1. Supplementary tables include a lot of symbols with no explanation what they stand for. Do they provide any more information than that provided already in the figures and their captions?

In the main figures of the manuscript, you can find the difference between the control and the rds group in each age and the difference between the different ages of the rds group.

Due to the enormous amount of data obtained when performing the two-way ANOVA, it was decided not to add the symbols that show the differences between the different ages of the control group to the figures. Instead, a supplementary table was made where the symbols that correspond to each of the ages of the control group are shown. To work with this table, you must first locate the stain you want, then the area you want to compare, and finally the age. Once the symbol that corresponds to that age has been located, its meaning can be found using Table 1.1 and it can be seen that other ages are different.

  1. There are some phrases with do not make sense: Abstract: “macromolecules oxidative damage” Starting in line 132-134: “The control and care of the mices … accomplished the Association for Research in Vision and Ophthalmology (ARVO) Statement for the Use of Animals in Ophthalmic and Vision Research.” Line 530-531: “inflammation ocular biomarkers”

We agree with reviewer 2 and we have corrected these mistakes in the original manuscript:

  • In line 14: “Expression of oxidative damage to macromolecules”
  • In lines 141-144: “The control and care of the mice were authorized by the CEU Cardenal Herrera Universities Committee for Animal Experiments (reference 2020/VSC/PEA/0094) and accomplished the Association for Research in Vision and Ophthalmology (ARVO) Statement for the Use of Animals in Ophthalmic and Vision Research.”
  • In lines 609-611: “The study of ocular oxidative stress and inflammation biomarkers can help us understand the physiopathology of RP and find a good strategy for monitoring the disease.”

In addition, further revisions by our group have been performed in order to find and correct other phrases.

  1. There are some spelling and other typographical errors. For example in lines 160, 161: “Deoxinucleotidyl” instead of “deoxynucleotidyl”

We agree the reviewer 2 and we have corrected the mistake in line 160 and 161. Now this sentence can be found in lines 181 and 182: “Terminal deoxynucleotidyl transferase (TUNEL) assay

To detect dying cells, a terminal deoxynucleotidyl transferase (TUNEL) assay was carried out with an in-situ detection kit, as stated in previous research (Roche Diagnostics, Mannheim, Germany) [31].”

Our group have reread the manuscript looking for misspelling mistake

Round 2

Reviewer 2 Report

The authors have addressed most comments and improved the manuscript except for the following:

Materials, Line 220: “Photos which have been taken at 20X magnification.” Was the magnification really only 20x? From images, it looks more like 200x.

1.      It is unclear why the quantification of GSH+GSSH and GFAP were done based on ratio area of fluorescence above selected threshold to the whole area instead of quantification of fluorescence intensity like for HNE.

2.      Methods: still not sufficient detail is given about the lighting conditions. LEDs can have various spectral outputs: usually there are cool white (with greater output in the short-wavelength range than the other two), daylight, and warm white and the colour temperature is given. Illuminance is useful for the human not for rodents, which have very different spectral sensitivity than the human eye. The irradiance should be provided together with the spectral characteristics of the light source.

3.      Avidin staining in the presence of biotin cannot be used for 8-OHdG quantification. This needs to be addressed in the text properly. The observed changes can be due to changes in biotin levels.  

4.      Some new phrases were introduced where the grammar needs to be corrected for the accuracy or clarity:

-        Abstract: “Expression oxidative damage to macromolecules, glutathione (GSH 14 and GSSG), GSH synthesis enzymes, glial fibrillar acidic protein (GFAP), ionized calcium binding 15 adapter molecule 1 (Iba1), and cluster of differentiation 68 (CD68) were studied”

-        Abstract: “ These findings contribute to understand that, in RP, oxidative stress and inflam-23 mation evolution and relation are dynamic time depending and highlight both processes as poten-24 tial therapeutic targets in this disease.”

-        Line 204: “macroglia expression (GFAP)”

5.      There are typographical errors in line 131-132

Author Response

ANSWERS TO REVIEWER 2, ROUND 2

We agree with the comments done by reviewer 2 and we appreciate the contribution to improve the quality of this paper.

Herein, we provide the comments detailing the changes have been made according to the reviewer suggestions.

Materials, Line 220: “Photos which have been taken at 20X magnification.” Was the magnification really only 20x? From the images, it looks more like 200x.

Regarding the magnification used in the quantifications, we have used photos taken at 20X, as it is explained in the material sections. With this magnification, the area examined of the retina is bigger.

However, in the figures, the photos used were taken at 40x (as stated in the figure legends) with the purpose of increasing the quality of the final figure and make able to the reader distinguish the details of the staining.  In some of the figures, it could be seen a white square and a digital magnification of this square in the side of the main photo with the same purpose.

We understand that this can lead to confusions and we have modified the sentence in the materials section to “Photos used for quantifications have been taken at 20X magnification and were analyzed using the software ImageJ Fiji version 1.52p (photos used in the figure of this manuscript have been taken at 40X magnification).” (lines 219 to 221).

  1. It is unclear why the quantification of GSH+GSSH and GFAP were done based on the ratio area of fluorescence above the selected threshold to the whole area instead of quantification of fluorescence intensity like for HNE.

The difference in the quantification method used for the GSH+GSSH and GFAP, and for the HNE is due to the non-specific area that labels the antibody. In the case of GSH+GSSH and GFAP as can be seen in figures 6 and 8 the labelling is specific and have a particular shape which could be easily distinguished from the background. In the case of HNE (figure 4), taking into account that is a product of oxidative stress over lipids, and lipids are present in the whole retina there is not any particular shape.

  1. Methods: still not sufficient detail is given about the lighting conditions. LEDs can have various spectral outputs: usually, there are cool white (with greater output in the short-wavelength range than the other two), daylight, and warm white and the colour temperature is given. Illuminance is useful for humans not rodents, which have very different spectral sensitivity than the human eye. The irradiance should be provided together with the spectral characteristics of the light source.

We agree with reviewer 2, and we have studied deeply this topic to be more precise and increase the accuracy of our paper. To provide an answer to the reviewer we have followed the guidelines in the paper: Peirson, S.N.; Brown, L.A.; Pothecary, C.A.; Benson, L.A.; Fisk, A.S. Light and the laboratory mouse. J. Neurosci. Methods 2018, 300, 26–36, doi:10.1016/j.jneumeth.2017.04.007.

Now in our manuscript this information is provided: “Light illuminance was provided by cool light led bulbs and the irradiance at the cage level was found to be between 6,22 to 9,32 µW/cm2.” (lines 138-139).

  1. Avidin staining in the presence of biotin cannot be used for 8-OHdG quantification. This needs to be addressed in the text properly. The observed changes can be due to changes in biotin levels.  

We agree with reviewer 2 that it is possible that the levels of biotin could interfere in the avidin labelling, and with that the result obtained. As the reviewer suggested we prevent the reader and this has been addressed in the text:.

The original sentence was:

“8-Oxoguanine is a major product of oxidative DNA damage. Interestingly, the structure of 8-oxoguanine is like the prevalent ligand for avidin [36]. In our study, avidin was used to recognize oxidatively injured DNA because other studies have established that staining of retinas with 8-oxoguanine antibody and avidin resulted in co-labeling of cells  in the ONL [36].”

Which has been substituted by:

“8-Oxoguanine is a major product of oxidative DNA damage. Interestingly, the structure of 8-oxoguanine is like the prevalent ligand for avidin [36]. In our study, avidin was used to try to recognize oxidatively injured DNA because other studies have established that staining of retinas with 8-oxoguanine antibody and avidin resulted in co-labelling of cells in the ONL [36]. Despite this fact, it should be taken into account that avidin stains also cells with high abundance of biotin and this can interfere in the results obtained. ” (lines 375 to 380).

  1. Some new phrases were introduced where the grammar needs to be corrected for accuracy or clarity:

We agree with reviewer 2 and we have changed those sentences in order to clarify and make them more understandable:

- Abstract: “Expression oxidative damage to macromolecules, glutathione (GSH and GSSG), GSH synthesis enzymes, glial fibrillar acidic protein (GFAP), ionized calcium binding 15 adapter molecule 1 (Iba1), and cluster of differentiation 68 (CD68) were studied”

has been substituted by:

“Oxidative damage to macromolecules, glutathione (GSH and GSSG), GSH synthesis enzymes, glial fibrillar acidic protein (GFAP), ionized calcium binding adapter molecule 1 (Iba1), and cluster of differentiation 68 (CD68) were studied” (lines 14 to 16).

-  Abstract: “These findings contribute to understand that, in RP, oxidative stress and inflammation evolution and relation are dynamic time depending and highlight both processes as potential therapeutic targets in this disease.”

has been substituted by:

“These findings contribute to understand that, in RP, oxidative stress and inflammation evolution and relation are time depending. In this sense, it is important to highlight that both processes are potential therapeutic targets in this disease.” (lines 22 to 25).

-        Line 204: “macroglia expression (GFAP)”

“macroglial reaction which is labelled by the marker GFAP” (now in line 203)

  1. There are typographical errors in line 131-132

We agree with reviewer 2 and we have changed the original sentence and now it can be read as follows:

“Control (C3Sn.BLiA-Pde6b+/DnJ, homozygous for Pde6b+) and rds mice (C3A.CG Pde6b+Prph2Rd2/J, homozygous for Prph2Rd2) were used in this work.” (lines 131 and 132).
